# The emergence of Gulf Stream and interior western boundary as key regions to constrain the future North Atlantic Carbon Uptake

Nadine Goris[1], Klaus Johannsen[2], and Jerry Tjiputra[1]

[1]NORCE Norwegian Research Centre, Bjerknes Centre for Climate Research, Bergen, Norway
[2]NORCE Norwegian Research Centre, Bergen, Norway

**Correspondence:** Nadine Goris (nadine.goris@norceresearch.no)

**Abstract.** In recent years, the growing number of available climate models and future scenarios has led to emergent constraints becoming a popular tool to constrain uncertain future projections. However, when emergent constraints are applied over large areas, it is unclear (i) if the well-performing models simulate the correct dynamics within the considered area, (ii) which key dynamical features the emerging constraint is stemming from, and (iii) if the observational uncertainty is low enough to allow for a considerable reduction of the projection uncertainties. We therefore propose to regionally optimize emergent relationships with the two-fold goal to (a) identify key model dynamics associated with the emergent constraint and model inconsistencies around them and (b) provide key areas where a narrow observational uncertainty is crucial for constraining future projections.

Here, we consider two previously established emergent constraints of the future carbon uptake in the North Atlantic (Goris et al., 2018). For the regional optimisation, we use a genetic algorithm and pre-define a suite of shapes and size-ranges for the desired regions. Independent on pre-defined shape and size-range, the genetic algorithm persistently identifies the Gulf Stream region centered around 30°N as optimal as well as the region associated with broad interior southward volume transport centered around 26°N. Close to and within our optimal regions, observational data of volume transport are available from the RAPID array with relative low observational uncertainty. Yet, our regionally optimised emergent constraints show that additional measures of specific biogeochemical variables along the array will fundamentally improve our estimates of the future carbon uptake in the North Atlantic. Moreover, our regionally optimised emergent constraints demonstrate that models that perform well for the upper ocean volume transport and related key biogeochemical properties do not necessarily reproduce the interior ocean volume transport well, leading to inconsistent gradients of key biogeochemical properties. This hampers the applicability of emergent constraints over large areas and highlights the need to additionally evaluate spatial model features.

## 1 Introduction and Motivation

At the heart of current investigations of the impact of possible future emissions pathways is the Coupled Model Intercomparison Project (CMIP). CMIP gathers the output of state-of-the-art climate models to a set of given experiments, designed to understand the drivers of climate change in a multi-model context. The CMIP-archive is commonly referred to in reports of the Intergovernmental Panel on Climate Change (e.g., IPCC, 2013, 2018) and has hence become fundamental for the creation of climate policies.

The first phase of CMIP, CMIP1, began in 1996 and included 21 global coupled climate models and a handful of experiments (Meehl et al., 1997, 2000). In contrast, the sixth and latest phase of CMIP (CMIP6, Eyring et al., 2016a) includes 312 experiments (Petrie et al., 2021) and anticipated output-data from at least 100 models hosted by 40 modelling centres (Balaji et al., 2018), though not every model participated in every experiment. Moreover, the model resolution has increased substantially over the years, additional Earth system processes and components have been introduced and an increased number of variables are required for each experiment (Petrie et al., 2021). Accordingly, the size of CMIP-data is increasing rapidly with a volume of 40TB related to CMIP3, 2PB for CMIP5 and an estimated 20PB for CMIP6 (Balaji et al., 2018).

Despite much progress in climate modelling, model bias and uncertainty (i.e., spread across models) have not decreased in many of the simulated variables. Most prominently, the model-generation of CMIP6 reveals the highest model uncertainty in equilibrium climate sensitivity when compared to other CMIP model-generations (Meehl et al., 2020). Similarly, Tagliabue et al. (2021) found that the absolute uncertainty in projections of global ocean net primary productivity has increased from CMIP5 to CMIP6. Additionally, their study points out that this growth in uncertainty substantially differs at regional scale. Contrarily, Terhaar et al. (2021) identify that the model uncertainty in surface density in the Arctic has decreased in CMIP6-ESMs when compared to CMIP5, leading to a reduced inter-model range of the anthropogenic carbon uptake in the Arctic. This result is echoed by Bourgeois et al. (2022), who find a smaller CMIP6 than CMIP5 model uncertainty in both the contemporary ocean stratification and the anthropogenic carbon uptake in the Southern Ocean between 30°S and 55°S. Yet, the combination of large data volume and partially high model uncertainty in CMIP6 makes a comprehensive evaluation of associated models and simulations highly challenging. Moreover, while observational estimates inform about present and past dynamics, it is often unclear how past and contemporary model biases affect their simulated climate change signal (Eyring et al., 2019). The emergent constraint approach (e.g., Hall et al., 2019) addresses this problem by identifying a relationship between observable characteristics of the current climate (predictor) and a certain aspect of future change (predictand) that emerge within a multi-model ensemble. Based on this relationship, it is possible to constrain the uncertainty of the model ensemble, assuming that a model's alignment with the observational estimate of the predictor is key to correctly simulate the predictand. Emergent constraints offer an attractive way of evaluating uncertain future projections. In the realm of Earth system projections, more than 50 emergent constraints have been found so far (Williamson et al., 2021). However, there are several concerns denoted when it comes to the usefulness of emergent constraints, including that a high cross-correlation between predictor and predictand can potentially reflect (i) the simplicity of a commonly used model parametrization and (ii) spurious relationships (Eyring et al., 2019). Hence, a physical explanation behind the emergent constraint is key for its plausibility (Williamson et al., 2021; Hall et al., 2019).

In ocean biogeochemistry, emergent constraints are often applied to variables that are averaged over large areas, as large scale ocean dynamics are crucial for many biogeochemical processes like the ocean carbon uptake (Kessler and Tjiputra, 2016; Goris et al., 2018; Bourgeois et al., 2022; Terhaar et al., 2021). Though these emergent constraints are physically plausible, we note that they deem a model to be the fittest due to its ability to simulate spatially averaged values of the predictor within observational uncertainty. There is no inspection if the models deemed to be 'fit' have a dynamically consistent predictor gradient within the considered region and we are not aware that this problem has been discussed yet. Yet, this is especially

relevant in cases where the predictor is closely linked to dynamical processes such as meridional advections. Moreover, the yielded constrained predictand is highly dependent on the observational estimate and a correct estimate of its uncertainty (Williamson et al., 2021). In the marine biogeochemical realm, in-situ observations are often too sparse in space and time to fully capture spatial and temporal variability, including fine-scale mixing, seasonal, interannual, decadal variability, long-term trends and short-term natural variability (Wang et al., 2019). Only few platforms reach the deep ocean, though its continuous observations are necessary e.g., to confidently capture the oceanic heat and carbon storage (Weller et al., 2019). The error occurring from the interpolation of sparse data is typically less well quantified than the observational error itself (Landschützer et al., 2020). Though the advent of biogeochemical ARGO floats gives the option to a substantial contribution to the goal of a 3-dimensional image of ocean biogeochemistry (Claustre et al., 2010), this potential is still far from being fully explored. While case-studies for selected regions exist (e.g., D'Ortenzio et al., 2020), estimates of observational uncertainty are often uncertain for emergent constraints in the realm of ocean biogeochemistry due to the large area covered by the emergent constraint and might hamper ongoing efforts to achieve a proper constraint for sensitivities of ocean biogeochemical variables. Due to these limitations of emergent constraints in the realm of biogeochemistry, our study is concerned with the regional optimization of emergent relationships with the two-fold goal to (a) identify key model dynamics for the emergent constraint and model inconsistencies around them and (b) provide key areas where a narrow observational uncertainty is crucial for constraining future projections. These key areas can be used to guide observational strategies.

In this study, we utilised two existing emergent constraints and applied a genetic algorithm to regionally optimise the area of the predictor, i. e., the observed variable. Our regional optimization explores different shapes and sizes of the sought-after area as an input and hence can be adapted for specific observational campaigns such as cruises. Moreover, the use of different shapes and sizes helps to identify key model dynamics for the emergent constraint and model inconsistencies around them. Both emergent constraints that we regionally optimise are related to the future carbon uptake of the North Atlantic and use (i) the seasonality of the oceanic partial pressure $pCO_2$ and (ii) the deep ocean storage of carbon since pre-industrial times as predictors (Goris et al., 2018). Both predictors could gain from an improved observational strategy as data is sparse on both seasonal timescales and in the deep ocean. Additionally, both predictors are highly dependent on the large scale ocean circulation such that it is of importance to not only study their averaged values over large areas but also the model performance within key regions and its dynamical consistency. We therefore consider this as the optimal test case for our regional optimisation. We note, however, that our study is primarily a showcase to illustrate the effectiveness of the genetic algorithm and to demonstrate the usefulness of regionally optimised emergent constraints. Our selection of the North Atlantic basin is motivated by its critical role for the long-term anthropogenic carbon sink and as the gateway to transport carbon from the surface to the deep ocean (Tjiputra et al., 2010). Further, the North Atlantic carbon uptake and dynamical features are relatively well studied (e.g., Olsen et al., 2008; Pérez et al., 2013; Goris et al., 2015) such that the plausibility of our results and their implications can readily be confirmed.

This paper is organised as follows: In Section 2, we introduce the concept of emergent constraints, the emergent constraints that we optimise as well as the genetic algorithm used for the regional optimisation and its experimental set up. When describing our results and discussing them in Section 3, we first describe the efficiency and performance of the genetic algorithm.

Subsequently, we present the optimal regions for both predictors as well as the associated regionally optimised emergent constraints and analyse the plausibility of our results as well as their implications. We discuss our approach and the additional information that it provides in Section 4. Our summary and conclusions can be found in Section 5.

## 2   Background and experimental set-up

### 2.1   The Emergent Constraint approach

The emergent constraint approach identifies an emerging quasi-linear relationship between characteristics of the current climate (predictor) and a certain aspect of future change (predictand) that emerge within a multi-model ensemble. Based on this relationship, it is assumed that models that simulate the predictor within observational uncertainties are better suited to simulate the predictand. Therefore, the emergent constraint approach utilises observations of the predictor to constrain the uncertainty around the simulated estimate of the predictand (e.g., Cox et al., 2013; Williamson et al., 2021). Our method of calculating

the constrained estimate follows the approach of Cox et al. (2013). Here, the unconstrained estimate of the predictand is given by the model mean and its uncertainty by the multi-model standard deviation. Assuming that all models are equally likely to simulate the true state of the predictand and are sampled from a Gaussian distribution, a probability density function (PDF) can be calculated for the unconstrained estimate using model mean and standard deviation. Similarly, PDFs of the observational estimate and of the linear regression between multi-model realisations of predictor and predictand are established. For the

observationally constrained predictand, a conditional PDF is calculated (Cox et al., 2013), i.e., a probability distribution of the predictand based on the established linear regression and under the condition that the predictor is within observational uncertainties. The observationally constrained estimate equals the mean value of the conditional PDF and the uncertainty of the estimate is given by its standard deviation. We note that emergent constraints come with a number of caveats, among them the fact that they are often applied over large areas and hence constrain a model's ability to simulate spatially averaged values

within observational uncertainty (see Section 1).

### 2.2   Emergent constraints of the North Atlantic future carbon uptake

As basis for our regional optimization, we utilise two emergent constraints that both constrain the future North Atlantic carbon uptake for an ensemble of 11 CMIP5-models under a high $CO_2$ future. Here, we give a short summary of these emergent constraints, for details the reader is referred to Goris et al. (2018). We note that the study of Goris et al. (2018) is concerned

with the "anthropogenically altered" component of the carbon cycle, defined as the outcome of either the historical scenario (for the years 1850-2005) or the high $CO_2$ future scenario (RCP8.5 experiment, years 2006-2100) minus that of the piControl experiment of the corresponding years. All variables calculated in this manner are henceforth marked by the subscript "ant*". $C_{ant*}$-uptake and $C_{ant*}$-storage can hence be equated to changes in oceanic carbon uptake and storage due to increasing atmospheric $CO_2$ and climate change.

Goris et al. (2018) found that their selected model-ensemble agrees fairly well on the North Atlantic $C_{ant*}$-uptake of the 1990s (defined as an average over the years 1990 to 1999), yet the simulated future North Atlantic $C_{ant*}$-uptake of the 2090s (defined as an average over the years 2090 to 2099) is highly uncertain. Here, some models simulate a future $C_{ant*}$-uptake of the same magnitude as that of the 1990s and other models project a future $C_{ant*}$-uptake that is 2-3 times higher than that of the 1990s (Fig. 1a). Goris et al. (2018) identified that discrepancies in the modelled North Atlantic future $C_{ant*}$-uptake arise

due to differences in the simulated efficiency of the high latitude transport of $C_{ant*}$-storage from the surface to the deep ocean. This transport is fuelled by deep mixed layer depths, high biological production and subsequent particle export to the deep as well as deep convection and subsequent interior ocean southward transport of $C_{ant*}$-storage out of the high latitudes.

    Two predictors associated with the contemporary efficiency of the surface-to-depth carbon transport were identified by Goris et al. (2018). The first predictor is the mid-to-high latitude summer (May to October) $pCO_2^{sea}$-anomaly of the 1990s, which is

135 tightly linked to winter mixing, nutrient supply and biological production, but also to deep convection (e.g., Olsen et al., 2008; Tjiputra et al., 2012). We note that Goris et al. (2018) utilised the negative mean summer $pCO_2^{sea}$-anomaly in order to be able to depict positive correlations. We follow this approach but opt to use the term mean 'winter $pCO_2^{sea}$-anomaly' (November to April) instead (Fig. 1b), defining it to be the deviation of the averaged winter $pCO_2^{sea}$-values from the mean annual $pCO_2^{sea}$-values and hence to equal the negative mean summer $pCO_2^{sea}$-anomaly. Goris et al. (2018) found that models with a low

future $C_{ant*}$-uptake have a negative contemporary mid-to-high latitude winter $pCO_2^{sea}$-anomaly. Their $pCO_2^{sea}$ seasonal cycle is driven by temperature meaning that their $C_{ant*}$-uptake is strongest in winter when surface temperatures are cold. Contrarily, models with a high future $C_{ant*}$-uptake have a positive contemporary mid-to-high latitude winter $pCO_2^{sea}$-anomaly, indicating that their seasonal cycle of $pCO_2^{sea}$ is dominated by variations of dissolved inorganic carbon (DIC) via biology and mixed layer depth. As the here considered models have differing timings for their peak in biological production (ranging from May to July)

and as seasonal warming and biological production are not in phase (the modelled peak in seasonal warming occurs in August), the highest correlations with the future North Atlantic $C_{ant*}$-uptake are yielded when the seasonal $pCO_2^{sea}$-anomaly covers the months from May to October (or November to March) and hence captures the different seasonal drivers at play. While both a DIC- and a temperature-driven $pCO_2^{sea}$ annual cycle leads approximately to the same contemporary $C_{ant*}$-uptake for the considered models, a temperature-driven $pCO_2^{sea}$ annual cycle leads to less $C_{ant*}$-uptake in the future due to ocean warming.

The second predictor is the fraction of the North Atlantic $C_{ant*}$ that is stored below 1000m depth (Fig. 1d), indicating how efficient the $C_{ant*}$-storage is transported into the deep ocean. Here, models that project a high future $C_{ant*}$-uptake have the majority of $C_{ant*}$-storage below 1000m depth, leading to a smaller fraction of $C_{ant*}$-storage at the surface and hence allowing for further $C_{ant*}$-uptake. For the second predictor, our analysis focuses on the time-frame 1997-2007 (hereinafter referred to as the 1997s) as the observation-based data-product, which we utilise to compare to the simulated fractional $C_{ant*}$-storage, is

normalized to the year 2002 (see Appendix A).

    By comparison to the observational database, these predictors allowed to constrain the model ensemble, and demonstrated that the models with more efficient surface-to-deep transport are best aligned with current observations (Fig. 1b,d). These models also show the largest future North Atlantic $C_{ant*}$-uptake, which hence appears to be the more plausible future evolution (Fig. 1c,e). We note that, within the selected model ensemble, the cross-correlation between the mid-to-high latitude winter

pCO$_2^{sea}$-anomaly of the 1990s and the future North Atlantic C$_{ant*}$-uptake is $r = 0.79$, while the cross-correlation between the fraction of the North Atlantic C$_{ant*}$ stored below 1000 m depth in the 1997s and the future North Atlantic C$_{ant*}$-uptake is $r = 0.94$. These correlations build the basis of the emergent constraint as they define how tight the relationships between predictors and predictand are across the model ensemble. We note that studies concerned with emergent constraints frequently use relationships with correlations lower than $r = 0.79$ (e.g., Qu et al., 2018; Selten et al., 2020; Mystakidis et al., 2017; Tokarska et al., 2020). Yet, the study of Goris et al. (2018) includes no regional optimization. Instead, it focuses on the broad surface areas of Mikaloff Fletcher et al. (2003) including the North Atlantic tropics (0.0°N to 17.781°N), low latitudes (17.781°N to 35.563°N), midlatitudes (35.563°N to 48.901°N), and high latitudes (48.901°N to 75.595°N) and the depth-boundary of 1000m depth as an indication for deep convection as well as for the horizon that separates between the upper and lower limbs of the Atlantic meridional overturning circulation (AMOC).

## 2.3 Experimental set-up for the regional optimisation

We apply a genetic algorithm (described in Section 2.4) to regionally optimise both predictors, i. e., to find a regionally condensed footprint of the already discovered relationship. This regional footprint might lead us even closer to the dynamical origin of the constraints and expose potential dynamical inconsistencies within the model ensemble but also allows to focus on smaller and more concentrated regions, which ultimately can be utilised for observational strategies and to refine observational uncertainties.

We consider the whole North Atlantic for regional optimization of the winter pCO$_2^{sea}$-anomaly predictor instead of focusing on the mid-to-high latitudes. Likewise, we consider all depth ranges of the fractional North Atlantic C$_{ant*}$-storage predictor for the optimisation, instead of focusing on the depth horizon below 1000 m depth. This way, the optimisation can confirm or reject the latitudinal boundaries and depth-ranges previously utlised by Goris et al. (2018). Before applying the regional optimisation, we re-gridded both the winter pCO$_2^{sea}$-anomaly and the fractional C$_{ant*}$-storage values from each model on a regular 1°x1° grid. We further interpolated the fractional C$_{ant*}$-storage on depth-levels at 100 m intervals. That way, it is more easily possible to construct new regions and apply them to the output of the whole model ensemble.

For our experimental set-up, we pre-defined the desired optimal regions in terms of geometrical shape. Specifically, we select two different shapes for both predictors, i.e. the winter pCO$_2^{sea}$-anomaly (2D-case) and the fractional C$_{ant*}$-storage (3D-case). For the 2D-case, the selected shapes are rectangles aligned with the longitudinal and latitudinal axes, respectively and arbitrary ellipses. For the 3D-case, we chose rectangular cuboids aligned with the longitudinal, latitudinal and depth axes, respectively and general ellipsoids. Our set-up of shapes is motivated by two criteria: (i) the possibility to capture regions of interest and (ii) to have a low dimensional search space allowing for a fast optimisation. The search space is of lower dimension for rectangles than for arbitrary ellipses and of lower dimension for cuboids than for general ellipsoids. Yet, arbitrary ellipses and general ellipsoids can be tilted within the surface water plane and the water volume, respectively, such that the associated optimal regions have the option to follow water masses more closely and are hence beneficial to consider. We note that other geometrical shapes would have satisfied both criteria. Among them is the option to optimize a tube, so that, for example, the ship track of an upcoming cruise can be optimised.

We additionally prescribed the approximate volume or area size that the optimal region should have. Here, we focus on areas and volumes of i) 10-20% , ii) 20-30% and iii) 30-40% of the total size of the North Atlantic surface area (for the 2D-case) or basin volume (for the 3D-case), respectively. In combination with rectangles, ellipses, cuboids and ellipsoids this results in 12 applications of the genetic algorithm. We note that the desired area can also be given as total area instead of a percentage and could, for example, be the distance that a cruise can cover within a given time-frame. Our choice of considering different area sizes is motivated by two considerations: firstly, we want to avoid spurious relationships, i.e., that the high correlation between the predictor spatially averaged over the optimal area and the predictand occurs without a direct causal relationship. If areas of different geometrical shapes and area sizes point towards the same key regions, it is less likely that the high correlations associated with these regions are non-causal, especially for diverse area sizes. Secondly, it is our goal to identify key model dynamics for the emergent constraint and model inconsistencies around them. A set of optimal areas of different shapes and geometrical forms allows us to inspect in more detail where key regions for the model performance are and if the simulated results for each of these regions are consistent with each other.

Apart from the size and shape limitations, we are also interested in solutions where the inter-model spread in the predictor is high as we want our regionally optimised emergent constraint to help us to constrain model spread. Therefore, we only consider grid-points within the optimal region where the multi-model standard deviation of the predictor is larger than the average multi-model standard deviation of the predictor for the whole North Atlantic.

## 2.4 Genetic algorithm and optimisation procedure

We utilize a genetic algorithm based on the algorithm described in Johannsen et al. (2022) to conduct the regional optimization of the predictors described in Section 2.2. Genetic algorithms are metaheuristics inspired by the process of natural selection that can be used to design flexible optimization algorithms. These algorithms go back to Holland (1975), who created genetic algorithms drawing on ideas from the field of biology. Since then, genetic algorithms have been developed by a growing community. The algorithms are increasingly popular due to their flexibility as they can be used in very general setting with non-differentiable or even discontinuous objective functions.

Genetic algorithms belong to the family of evolutionary algorithms and are inspired by Darwinian evolution (Sivanandam and Deepa, 2007). They mimic natural evolution through reproduction, mutation and selection to find (close-to) optimal solutions for highly complex problems. Constitutive elements of genetic algorithms are a population formed by a number of individuals (characterized by genes and equipped with phenotypical expressions and fitness), selection of parents and reproduction (creation of offspring), mutation and selection of surviving individuals (survival is determined via fitness). Subsequently, the original population is replaced by the surviving individuals, forming the next generation. For a number of generations, the steps outlined above are repeated. In this way, the algorithm can approximate the (close-to) optimal solution, determined by the fittest individual over all generations. In our case, individuals correspond to domains and the fittest individual is the domain for which spatially-averaged values of the predictor reach the highest correlations when correlated with the predictand. In the following paragraphs, we describe our set-up of a genetic algorithm to perform the regional optimization of this study.

As we regionally optimise the predictors of two emergent constraints and utilise two pre-defined area/volume shapes and three pre-defined area/volume size-ranges for each predictor (see Section 2.3), our genetic algorithm is applied 12 times to a population of individuals. We utilise 4 different types of individuals, that is rectangles, ellipses, cuboids and ellipsoids (see Section 2.3). Genetic algorithms express an individual as a specific combination of genes. Here, we express a rectangle as four continuous genes, where the first and second genes describe the south-western point and the third and fourth genes describe the north-eastern point of the rectangle in longitude-latitude coordinates (Fig. 2b). An ellipse is described by five genes (Fig. 2b), consisting of a shift vector (two genes) and a symmetric positive definite matrix (encoded by three genes). The shift vector is the center of the ellipse and the eigenvectors of the symmetric positive definite matrix are the principal axis of the ellipse. A cuboid is encoded by six genes (Fig. 2b). The first three genes describe the south-western point at the shallowest ocean depth (longitude, latitude and depth) and the forth to sixth genes describe the north-eastern point at the deepest ocean depth (longitude, latitude and depth). Similar to the ellipse, an ellipsoid (Fig. 2b) is described by a shift vector (three genes) and a symmetric positive definite matrix (six genes). The shift vector is the center of the ellipsoid and the eigenvectors of the symmetric positive definite matrix are the principal axis of the ellipsoid.

In order to find the fittest individual or the optimal domain, the genetic algorithm maximises a fitness function. In our study, the first part of the fitness function is the cross-correlation between (i) the simulated predictor-values per model (contemporary winter $pCO_2^{sea}$-anomaly or contemporary fractional $C_{ant*}$-storage) averaged over the region specified by the considered individual and (ii) the simulated predictand per model (the future $C_{ant*}$-uptake of the North Atlantic). The cross-correlation describes how tight the relationship between predictor and predictand is and higher values corresponds to a higher fitness for an individual. As we additionally prescribed the approximate volume or area size of the optimal region, our fitness function includes a penalization to ensure compliance with the area or volume-condition. If an area or volume is not compliant with the size condition, a negative value smaller than -1, which is decreasing with the area or volume-violation, is added.

For each of our applications of the genetic algorithm, we use a population of 1000 individuals evolving over 100 generations. As initialization, a population of (i) 1000 rectangles or ellipses of varying area-sizes are placed randomly across the surface of the North Atlantic (2D-case) or of (ii) 1000 cuboids or ellipsoids of varying volume-sizes are placed randomly across the water volume of the North Atlantic (3D-case). Subsequently, each individual gets a fitness assigned based on our fitness-function. After our initialisation, we create a new generation by applying three steps (see Fig. 2a): (1) A new population of 1000 individuals is created through a repeated tournament selection. In the tournament selection process, 10 individuals are selected at random and the fittest of these is chosen (Eiben and Smith, 2003). This process is repeated 1000 times to create the new population. We note that the resulting population in general contains a number of identical individuals. (2) We randomly chose 50% of the individuals of our new population (this equals a crossover probability of $p = 0.5$) as parents, create two offspring for each pair of parents and use the offspring to replace their parents. This leads to a population of 500 selected individuals and 500 offspring. To create an offspring, we use a one point crossover with random position (see Fig. 2c), i. e., within the sequence of genes of both parents, a crossover site is selected at random. If, for example, an individual is defined by 4 genes (as this is the case for the rectangle, illustrated in Fig. 2c) and the crossover site is between the first and the second gene, then the first gene of one offspring will be defined by one parent, while the second to fourth gene is defined by the other

parent (Sastry et al., 2005). (3) We mutate 20% of the revised population (this equals a mutation probability of $p = 0.2$) and replace the corresponding individuals with their mutations. We realise mutation using Gaussian mutation, where a vector of Gaussian noise is added to the vector of genes (Kramer, 2017). As Gaussian noise we choose a mean of zero and a standard deviation of 0.05. After these three steps, we have a new generation consistent of selected copies, selected and mutated copies, offspring and mutated offspring. Subsequently, the fitness of each individual of our new generation is evaluated. The assigned fitness is then utilised for another iteration of the genetic algorithm via repetition of steps (1)-(3). This algorithmic sequence is illustrated in Fig. 2a. For the purpose of our study, we fix the number of iterations to 100 and stop the algorithm afterwards. The fittest individual of all generations is then defined to be our (close-to) optimal solution.

## 3 Results

In this Section, we first describe the (close-to) optimal cross-correlations obtained through regional optimisation of our emergent constraints as well as the associated speed of convergence of the genetic algorithm (Section 3.1). For both predictors, we separately illustrate the optimal regions and their related emergent constraints (Sections 3.2/ 3.3), their plausibility (Sections 3.2.1/ 3.3.1) and their implications for both models dynamics and observational strategies (Sections 3.2.2/ 3.3.2).

### 3.1 Towards an optimal solution in 100 iterations

Cross-correlations between the simulated values of the future North Atlantic $C_{ant*}$-uptake and values of both predictors within the optimal regions identified by the genetic algorithm are significantly improved as compared to the original emergent constraints (see Fig. 3). In the 2D-case, the original cross-correlation of $r = 0.79$ is improved to $r = 0.863$, $r = 0.855$ and $r = 0.848$ for the rectangle solutions with 10-20%, 20-30% and 30-40% of North Atlantic surface area, respectively and $r = 0.863$, $r = 0.856$ and $r = 0.852$ for the ellipse solutions with corresponding area sizes. For the 3D-case, the already high original cross-correlation of $r = 0.94$ is still improved to $r = 0.972$, $r = 0.966$ and $r = 0.966$ for the cuboid solutions with 10-20%, 20-30% and 30-40% of North Atlantic volume size, respectively, and $r = 0.987$, $r = 0.975$ and $r = 0.970$ for the ellipsoid solutions with corresponding volume sizes. We note that, in general, higher cross-correlations are achieved for smaller areas or volumes due to more placement possibilities. While this is not surprising, this might lead to the desire to use shapes that are even smaller than our predefined volume- and area-limits. For our application, however, we advise against using shapes of very limited volume. This is based on the fact that we are searching for areas that provide a fingerprint of the original emergent constraints for the North Atlantic future $C_{ant*}$-uptake. Here, the original emergent constraints are based on features that are associated with the large scale ocean circulation. While the algorithm would be able to find high cross-correlations for shapes of smaller size, it would be difficult to assign the outcome to those large-scale circulation features and to assign a dynamical interpretation to the so-obtained optimal regions.

Regarding the speed of convergence for the 2D-case, the first iteration of the genetic algorithm already reaches cross-correlation of $r = 0.863$, $r = 0.854$ and $r = 0.847$ for the rectangle solution with area-sizes of 10-20%, 20-30% and 30-40%, respectively and only offers improvements in the fourth decimal point afterwards (Fig. 3a). After four iterations, there is

subsequently no improvement in the first 10 decimal points. For the ellipse solutions, the first iteration yields cross-correlations of $r=0.857$, $r=0.849$ and $r=0.851$ for area-sizes of 10-20%, 20-30% and 30-40% and only improvements in the third or subsequent decimal points are subsequently achieved. In contrast to the rectangle solutions, the genetic algorithm converges slower for the ellipse solutions (Fig. 3b) and, for the area-size of 30-40%, no subsequent improvement in the first 10 decimal points is only reached after the 66th iteration. In general, for the 2D-case, there is a fast speed of convergence that can be traced back to the limited area that the genetic algorithm operates in and the associated limited options for placement.

Compared to the convergence of the rectangle solutions, the convergence of the cuboid-solutions is a bit slower between the first and tenth iteration due to more placement options throughout the water column (Fig. 3c). Yet, the first iteration of the cuboid-application of the genetic algorithm reaches already cross-correlations of $r=0.962$, $r=0.964$ and $r=0.962$ for 10-20%, 20-30% and 30-40% of North Atlantic volume size. Subsequently, only improvements in the third decimal place are achieved and after 10 iterations there is no improvement in the first 10 decimal points.

In contrast to the cuboid solutions and all applications of the 2D-case, all ellipsoid solutions show a slightly different convergence-behaviour (Fig. 3d). Here, the cross-correlations are still significantly increasing at the end of our application of the genetic algorithm. At the same time, the maximum cross-correlation of the smaller ellipsoids during our execution of 100 iterations are 0.015 and 0.009 higher than those of the smaller cuboids. We assign both the slow speed of convergence as well as the improved cross-correlations of the smaller ellipsoid to the higher degrees of freedom as well as to more placement options as the smaller-volume ellipsoids have the option to be tilted within the water column.

### 3.2 Optimal regions for the winter pCO$_2^{sea}$-anomaly and associated new emergent constraints

The optimal regions found by the genetic algorithm for the winter pCO$_2^{sea}$-anomaly (2D-case) all have their southern boundary at 28°N or 29°N, independent on predefined shape and size (Fig. 4). Their northern boundaries vary between 43°N and 53°N, with larger optimal areas reaching further north. Longitude-wise, all optimal areas are placed in the western part of the North Atlantic. Here, their western and eastern boundaries vary dependent on the predefined size-range and shape of the optimal area. Yet, the area between 73°W and 30°W and between 29°N and 42°N is enclosed by all optimal areas and is hence central for the considered emergent constraint. This central area is very similar to the optimal rectangle and ellipse covering 10-20% of the North Atlantic area size, which yield the highest cross-correlations when compared to the optimal rectangles and ellipses with larger surfaces, respectively (see Section 3.1). We note that, for the optimal areas and their given size-requirements, a placement further south than 28°N was not possible as only grid-points where the multi-model standard deviation of the predictor is larger than that of the mean multi-model standard deviation of the predictor of the North Atlantic are eligible for our optimal regions (see Section 2.4). It can be readily seen in Fig. 4b-d that our requirements for eligible grid points excludes the lower latitudes of the North Atlantic from being chosen for placement of the optimal region.

We utilise the optimal regions to spatially average the winter pCO$_2^{sea}$-anomaly over each of them individually and constrain our predictand (Fig. 5). For details of the method that we utilise to calculate the unconstrained and observationally constrained estimates of the future North Atlantic C$_{ant*}$-uptake, the reader is referred to Section 2.1. For the 2D-case, the unconstrained estimate of our model ensemble yields a mean value of 0.5±0.23 PgC/yr for the future North Atlantic C$_{ant*}$ uptake, while

the original emergent constraint corrected this to $0.73 \pm 0.27$ PgC/yr. When applying our regionally optimised predictors, the observational constraints correct the unconstrained values towards mean values between 0.72 PgC/yr and 0.79 PgC/yr (Fig. 5,

Table 1).

As outlined in Section 1, our regional optimization of emergent relationships has the two-fold goal to (a) identify key model dynamics for the emergent constraint and model inconsistencies around them and (b) provide key areas where a narrow observational uncertainty is crucial for constraining future projections. However, before following up with our goal, we need to ensure that there is a physical explanation behind the optimal areas found as this is key for the plausibility of emergent

constraints (Williamson et al., 2021; Hall et al., 2019). Therefore, we utilise Section 3.2.1 to investigate the plausibility of the optimal areas found before examining our two-fold goal in Section 3.2.2. We note however, that our investigation of the plausibility of the optimal regions is closely related to model dynamics and hence to part of our goal.

### 3.2.1 Plausibility of the optimal areas for the winter $pCO_2^{sea}$-anomaly

As all of our six optimal areas cover the same central area between $73°W$ and $30°W$ and between $29°N$ and $42°N$, we consider

it less likely that the high correlations between the predictor spatially averaged over the optimal areas and the predictand are spurious. Therefore, we proceed to investigate the physical explanation of the identified optimal domains. For our optimal regions, the simulated differences in the winter $pCO_2^{sea}$-anomaly are especially well related to a model's future North Atlantic $C_{ant*}$-uptake. We expect large scale circulation features to be an important driver of model differences in the winter $pCO_2^{sea}$-anomaly as these are directly related to nutrient supply, heat transport and deep mixing. Based on this logic, the identified

optimal regions seem reasonable as they all cover a major part of the Gulf Stream. The Gulf Stream is a key part of the warm and upper branch of the Atlantic Meridional Overturning Circulation (AMOC), which transports waters from the low-latitude North Atlantic via the Gulf Stream, the North Atlantic Current (NAC) and the Irminger Current to the high latitude North Atlantic, thereby releasing heat to the atmosphere (e.g., Rhein et al., 2011). Along this path, deep mixed layers are formed via wind-driven velocity shears but also via heat-loss to the atmosphere, which becomes more prominent in higher latitudes

where it leads to deep convection (e.g. Rhein et al., 2011). The strength of the Gulf Stream and its extension is not only an important driver of the amount of heat that is transported from low to high latitudes and the strength of deep convection in high latitudes but also for transporting high-nutrient thermocline waters from low to high latitudes (the so-called nutrient-stream, see e.g., Williams et al., 2011) and hence for the strength of the winter $pCO_2^{sea}$-anomaly. In line with this, the model spread is increasing further north and the highest multi-model standard deviation of the contemporary winter $pCO_2^{sea}$-anomaly (Fig. 4b)

follows the path of the NAC, which is the immediate Gulf Stream extension.

In first instance, it seems surprising that not all optimal regions cover the path of this high standard-deviation, but that the smallest optimal regions are placed directly at the southwestern boundary of it, which coincides with the beginning of the Gulf Stream. However, we note that high multi-model standard deviations might also indicate a slightly different placement of currents between models and that the paths of Gulf Stream and NAC in the open ocean are influenced by decadal variations,

which might not be in phase within the model ensemble. The optimal regions cover those latitudes before and where the Gulf Stream starts to separate from the coast and where the spatial path of the current is therefore less variable within models.

Additionally, we note that this placement seems reasonable as biological production becomes more dominant further north. Here, different ecosystem model-parametrisations get a larger imprint on the simulated contemporary winter $pCO_2^{sea}$-anomaly, such that the cross-correlations between predictor and predictand are not only based on surface temperature, available nutrients and mixed layer depth.

We use further calculations to support our plausibility-argument that our optimal regions capture the influence of the upper branch of the AMOC, specifically the Gulf Stream, on the simulated contemporary winter $pCO_2^{sea}$-anomaly and hence on our predictand, the future North Atlantic $C_{ant*}$-uptake. For this, we calculate cross-correlations between our predictand and the simulated strength of the upper AMOC branch (see Appendix B) at $30°$N, as this is a central latitude in our identified optimal regions. As we consider the AMOC volume transport only in terms of driving the contemporary winter $pCO_2^{sea}$-anomaly, we expect the transport within the mixed layer to be key. Indeed, when calculating cross-correlations between 10-year running averages of the accumulated northward volume transport between surface and different depths at $30°$N and our predictand, we identify cross-correlations to be highest for the accumulated northward volume transport between surface and 500 m. The cross-correlations get worse for both shallower and deeper depths when varying the lower boundary of the northward volume transport in depth-intervals of 100 m (Fig. 6a). We note that cross-correlations between 10-year running averages of our predictand and the northward volume transport between surface and 500 m at $30°$N stay between $r=0.845$ and $r=0.921$ for all considered time-periods (Fig. 6a), with a cross-correlation of $r=0.883$ for the 1990s. This value is slightly above the cross-correlations between the modelled contemporary winter $pCO_2^{sea}$-anomaly in our optimal regions and the predictand.

In order to quantify that these high cross-correlations between our predictand and the accumulated northward volume transport between surface and 500 m are a specific feature of our identified optimal regions, i.e. the Gulf Stream region, we further vary the latitude of the northward volume transport in our calculations in latitude-intervals of $5°$ (Fig. 6b). When utilising 10-year running averages of the northward volume transport between surface and 500 m, we find that cross-correlations are highest at $25°$N or $30°$N, depending on the year considered, and that cross-correlations get worse for latitudes further north and south. Specifically between $30°$N and $35°$N, the cross-correlations are decreasing rapidly. For most of the considered decades, cross-correlations are slightly higher at $25°$N than at $30°$N. However, this latitudinal band contains no eligible grid-points in the Gulf Stream region, so that the genetic algorithm could not identify it to be part of an optimal region. We conclude that it is indeed in the Gulf Stream region where cross-correlations between our predictor and the predictand are exceptionally high. We deem the identified optimal regions to be characteristic of the northward volume transport of a model, governing its surface temperature distribution, available nutrients and mixed layer depths not only at the specified latitudes of the optimal regions, but along the path of the Gulf Stream, NAC and Irminger currents from low-to-high latitudes. This confirms the plausibility of our optimal regions for the contemporary winter $pCO_2^{sea}$-anomaly.

We would like to additionally denote that the relationship between maximum AMOC-strengths and the North Atlantic carbon sink might not be as strong as commonly assumed in modelling studies. Cross-correlations between 10-year running averages of the maximum northward volume transport at our central latitude of $30°$N or the commonly used latitude of $40°$N and our predictand are between $r=0.652$ and $r=0.870$ and between $r=0.575$ and $r=0.790$ for all considered time-periods, respectively. Both relationships associated with maximum AMOC-strength yield weaker correlations with the future North

Atlantic $C_{ant*}$-uptake than the northward volume transport within the mixed layer. When studying the North Atlantic carbon sink, we hence propose to rather focus on the northward volume transport within the mixed layer at latitudes between 25°N and 30°N.

### 3.2.2 Implications of the optimal areas of the winter $pCO_2^{sea}$-anomaly

After having verified the plausibility of the optimal areas of the winter $pCO_2^{sea}$-anomaly, we follow up on the two-fold goal of our regional optimization of emergent relationships. Our optimal areas directly fulfill one part of our goal by indicating key areas where a narrow observational uncertainty is crucial for constraining future projections. With regards to our second goal of identifying key model dynamics for the emergent constraint, our plausibility-analysis identified the northward volume transport of a model to be the key driver of the emergent constraint between the winter $pCO_2^{sea}$-anomaly and future North Atlantic $C_{ant*}$-uptake, via governing its distributions of temperature, available nutrients and mixed layer depths from low-to-high latitudes. Based on this, we examine the emergent constraints of our regionally optimised winter $pCO_2^{sea}$-anomaly (Fig. 5) for model inconsistencies around these key model dynamics.

We find that all newly obtained constrained values for the future North Atlantic $C_{ant*}$ uptake are consistent with each other, i.e., the differences in the mean values are small and the uncertainties around the mean values ensure that the solutions do not contradict each other (Fig. 5, Table 1). Nevertheless, the constrained mean values of the future North Atlantic $C_{ant*}$ uptake based on the smallest optimal ellipse or rectangle are consistently smaller than those based on the largest optimal ellipse or rectangle (Fig. 5), which reach further north (Fig. 4c,d). Similarly, areas positioned further south (Fig. 5a,c) generally have models with lower future North Atlantic $C_{ant*}$ uptake closer to their mean observational value of the $pCO_2^{sea}$-anomaly than those positioned further north (Fig. 5b,d), equal to the observational constraint shifting further right within the model ensemble (from Fig. 5a,c to Fig. 5b,d). Between the smallest and the largest rectangles or ellipses, the observational mean value of the winter $pCO_2^{sea}$-anomaly increases by 5.85 $\mu$atm or 7.18 $\mu$atm, respectively, while the average value of the winter $pCO_2^{sea}$-anomaly of the four models that are within observational uncertainty for all optimal areas only increased by 1.89 $\mu$atm or 3.99 $\mu$atm, respectively. The seven remaining models show an even smaller increase of 0.01 $\mu$atm or even a decrease of 0.73 $\mu$atm, respectively. This could indicate that the south-north gradient of the winter $pCO_2^{sea}$-anomaly is not steep enough in the model ensemble, i.e., that the modelled northwards-propagation of related properties is too weak (this relationship is visualised for the winter $pCO_2^{sea}$-anomaly gradient between the southernmost and northernmost latitudes of the smallest rectangle in supplementary Fig. S2). However, the uncertainties around the observational estimates of the winter $pCO_2^{sea}$-anomaly are large and do not allow us to be certain about the observed south-north gradient and hence potential discrepancies in the modelled south-north gradient. We use observational estimates of the upper (0-500 m) North Atlantic northward volume transport to further investigate a potentially too weak northward propagation (confirmed to be a plausible predictor in Section 3.2.1), due to limited observational availability only considered at 26.5°N and for the time period 2005-2014 (see Appendix A). The transport values show that the northward propagation of the seven models with the lowest future $C_{ant*}$ uptake is notably too weak, but that the upper ocean northward transport of the four models with the highest future $C_{ant*}$ uptake is within observational uncertainties (see supplementary Fig. S1). Yet, the model ensemble shows diverse changes of this transport between 26°N

and the latitudes of the optimal areas. Here, the transport of the four models with the highest future $C_{ant*}$ uptake shows an average increase of 1.86 Sv between 26°N and 30°N, and we find an average increase of 0.65 Sv for the remaining 7 models. Without an additional observational estimate at 30°N (or another latitude of the optimal areas), we cannot confirm or deny if the northward propagation of the four best models is within observational bounds for our optimal regions.

### 3.3  Optimal regions for the fractional $C_{ant*}$-storage and associated new emergent constraints

In the case of cuboids-solutions, all optimal areas identified by the genetic algorithm for the contemporary fractional North Atlantic $C_{ant*}$-storage (3D-case) are placed in the western part of the North Atlantic (Fig. 7c) with a common western boundary at 96°W and southern boundaries at 19°N (smallest cuboid) or 18°N (larger cuboids). Their northern and eastern boundaries vary between 34°N and 50°N as well as 61°W and 31°W, respectively, with larger cuboids reaching both further north and east. With the given size-requirements and and the grid point eligibility criterion (see Section 2.3), a placement of the optimal cuboids further south is unlikely. We note that the eligibility of grid-points is considered per depth-layer, such that the illustrated depth-integrated values of the multi-model standard deviation (Fig. 7b) only give a first indication of eligible points (non-eligible points are visualised per depth layer in supplementary Figs. S5 and S6). The genetic algorithm identified the optimal depth-ranges for the cuboids to be 700 - 4700 m for the smallest cuboid as well as 800 - 4900 m for the larger cuboids. Apart from the depth-range of 700 - 800 m, the optimal cuboids of larger volumes are enclosing the optimal cuboids of smaller volumes. As the cross-correlations between the simulated future North Atlantic $C_{ant*}$-uptake and the fractional $C_{ant*}$-storage within the optimal cuboids is also highest for the smallest cuboid (see Section 3.1), we consider its enclosed volume to be central for our emergent constraint.

The optimal depth-ranges identified by the genetic algorithm for the ellipsoids are 0 - 4800 m for the smallest and 0 - 5000 m for the medium-sized and the largest ellipsoid. The surface positions of the vertical principal axis of the smallest and the medium-sized ellipsoids are in the eastern North Atlantic around 25°W/ 40°N and they tilt in south-west direction with depth until being positioned in the western North Atlantic at around 75°W/ 25°N for their deepest points (Fig. 7d-e). Contrarily, the vertical principal axes of the largest ellipsoid tilts in direction north-east with depth and its position is already in the western North Atlantic for its shallowest point (Fig. 7f).

To constrain our predictand, we spatially average the fractional $C_{ant*}$-storage over each of our optimal regions (Fig. 8). Our regionally optimised predictors lead to associated constrained estimates of the future North Atlantic $C_{ant*}$ uptake with mean values between 0.55 PgC/yr and 0.79 PgC/yr (Fig. 8, Table 1). In comparison, the unconstrained estimate of the future North Atlantic $C_{ant*}$ uptake is 0.5 ± 0.23 PgC/yr, and the original emergent constraint corrected this to 0.64 ± 0.26 PgC/yr.

Before we follow up on our two-fold goal associated with the regional optimization of this emergent relationship, we follow the same approach as for the 2D-case and investigate the plausibility of the optimal areas found first (Section 3.3.1) and only subsequently report on our two-fold goal (Section 3.3.2). However, there is a close relation between our investigation of the plausibility of the optimal regions and our goal of identifying key model dynamics for the emergent constraint.

### 3.3.1 Plausibility of the optimal areas for the fractional $C_{ant*}$-storage

We find that all three optimal cuboids cover the same central area between 96°W and 61°W, 19°N and 34°N and 800 m and 4700 m, such that the high correlation between the predictor spatially averaged over the optimal cuboids and the predictand are less likely to be spurious. Similarly, all optimal ellipsoids appear to cover the relatively slow and broad interior pathway west of the North Atlantic for ocean depths below 1000 m, yet the similarity of the optimal ellipsoids is more difficult to establish. To confirm the plausibility of our optimal areas, we hence investigate the physical explanation for the regionally optimised emergent constraint.

Our identified optimal regions for the predictor point out key regions, where the simulated differences in the fractional North Atlantic $C_{ant*}$-storage are especially well related to a model's future North Atlantic $C_{ant*}$-uptake. The contemporary fractional North Atlantic $C_{ant*}$-storage is a measure for the efficiency of carbon sequestration (Goris et al., 2018), which reflects not only the strength of high latitude deep convection and sinking organic particles, but also of southward volume transport of $C_{ant*}$ in deeper ocean depths. This feature is tightly related to our predictand, the future North Atlantic $C_{ant*}$-uptake, as the pathways of carbon sequestration ultimately determine how much $C_{ant*}$-storage is efficiently removed from the high latitude North Atlantic ocean surface and hence how much $C_{ant*}$ can subsequently be taken up across the air-sea interface. Here, a more efficient carbon sequestration, i. e. less storage of $C_{ant*}$ in shallower depths and more storage in the deeper ocean leads to the potential for more $C_{ant*}$ uptake in a high $CO_2$ future. We hence expect our optimal regions to be a reflection of important carbon sequestration pathways.

As the ellipsoids can be tilted within the water volume, the associated optimal regions have the option to follow water masses more closely. Their optimal solutions allow us to visually quantify if the reasoning of the predictor being a measure of pathways of carbon sequestration (Goris et al., 2018) holds. While the placements of the optimal ellipsoids in shallower ocean layers are still influenced by mixed layer dynamics and the pathways of carbon sequestration are difficult to identify, the optimal ellipsoids are placed in central areas of the simulated fractional $C_{ant*}$-storage pathways for deeper layers. We note that the spatial gradients of the fractional $C_{ant*}$-storage multi-model mean (displayed for different depth in supplementary Figs. S3 and S4) are consistent with the theory that the deeper and southward branch of the North Atlantic volume transport can be divided into (i) a fast and narrow boundary pathway and (ii) a relatively slow and broad interior pathway west of the North Atlantic ridge (Gary et al., 2011, and references therein). However, the multi-model standard deviation of the fractional $C_{ant*}$-storage as displayed in Fig. 7b (and additionally displayed for different depths in supplementary Figs. S5 and S6) indicates that the models do not agree on the strength of this southward transport, neither for its slow nor for its fast component. For ocean depths below 1000 m, the optimal ellipsoids consistently point towards the areas of the relatively slow and broad interior pathway west of the North Atlantic ridge with both high fractional $C_{ant*}$-storage multi-model mean values and standard deviations. We hence consider the optimal ellipsoids to be in accordance with the previous reasoning of Goris et al. (2018), though we note that it was difficult to relate the ellipsoid-shapes to physical meaning.

The cuboid-solutions are implemented in a way that prevents them from being tilted within the water volume and they hence can not follow the $C_{ant*}$ sequestration pathway as closely as the ellipsoid solutions. All optimal cuboids seem to point roughly

towards the southernmost points that the relatively slow and broad interior southward transport of $C_{ant*}$ reaches to, though the narrow and fast southward transport of $C_{ant*}$ reaches further south (both are indicated through the horizontal gradient in the fractional $C_{ant*}$-storage multi-model mean as illustrated in Fig. 7b and supplementary Figs. S3 and S4). This placement seems to support our argument that the optimal cuboids-solutions capture the influence of the transport pathways of the carbon sequestration.

As previously done in the 2D-case, we support our argument with respect to the optimal cuboids with additional calculations. Here, we calculate cross-correlations between our predictand and the streamfunction volume transport at 26°N (see Appendix B), as this is the latitudinal mid-point of the smallest cuboid and hence a central latitude of our identified optimal cuboids. To validate the depth-boundaries identified by the smallest and central cuboid, we set one boundary of the volume transport to be one of the identified depth-boundaries of the cuboid, while we vary the other depth-boundary (Fig. 9a,b). Cross-correlations between 10-year running averages of the accumulated volume transport in different depth-ranges at 26°N and our predictand show that cross-correlations are highest for the accumulated southward volume transport between 900 - 4700 m when varying the upper depth boundary (Fig. 9a) and between 700 - 5300 m (or even deeper) when varying the lower depth boundary (Fig. 9b). While this seems to indicate that the depth-boundaries of the cuboids are not optimal, we note that the cross-correlations obtained for upper depth boundaries of 900 m and 700 m are relatively similar and a strong decline in cross-correlations only appears for an upper depth boundary above 500 m. Moreover, the $C_{ant*}$ southward transport is strongly influenced by the amount of $C_{ant*}$ that is available for transport in a specific depth-layer and while the lower depth boundary of 5300 m reaches higher cross-correlations between 10-year running averages of the southward volume transport and the predictand, the amount of $C_{ant*}$ that can be transported in these deep depth layers is negligible. Additionally, there are no eligible grid-points in these very deep layers.

When considering the streamfunction volume transport within the depth-boundaries given by the smallest optimal cuboid and varying its latitudes (Fig. 9c), we find that the 10-year running averages of volume transport at the identified mid-latitude of the smallest cuboid offers significantly higher cross-correlations with our predictand than the volume transport further north. This points towards the optimal cuboids capturing an important latitude of the southward interior $C_{ant*}$-transport. However, the volume transport south of the cuboid's placements offers slightly higher cross-correlations with the predictand. Yet, in these latitudes south of our cuboids, the amount of deep $C_{ant*}$-storage available for southward transport is small and there are moreover very few eligible grid-points in these latitudes. Under the conditions given to the genetic algorithm, the identified depth ranges and latitudes hence seem plausible. Cross-correlations between our predictand and 10-year running averages of the southward volume transport at the identified depth-ranges and latitudinal mid-point of the smallest cuboid are between $r = 0.690$ and $r = 0.859$ for all time-periods and $r = 0.771$ for the analysed time-period 1997-2007. The identified cross-correlations indicate a strong link between southward volume transport and our predictand, and hence verify its plausibility. We note, however, that the fractional $C_{ant*}$-storage offers an better relationship with our predictand than the southward volume transport. This comes as no surprise as the depth-distribution of the $C_{ant*}$-storage plays a big role in its southward transport.

### 3.3.2 Implications based on the optimal areas of the fractional $C_{ant*}$-storage

In order to get a more certain estimate of the future North Atlantic $C_{ant*}$-uptake, a reduction of the observational uncertainty of the $C_{ant*}$-storage within out optimal areas would be highly valuable. Yet, it might be operationally more challenging to encompass the optimal ellipsoids during a cruise, while the optimal cuboids might be represented with observations more easily.

Our optimised emergent constraints can moreover inform us about model inconsistencies within key dynamical features. For the smallest and medium-sized ellipsoids, the observational uncertainty does not allow for constraining the solution further (see Fig. 8c for the smallest ellipsoid). For the largest ellipsoid and the optimal cuboids, we find that all newly obtained constrained values for the future North Atlantic $C_{ant*}$ uptake are consistent with each other, i.e. the solutions do not contradict each other (Fig. 8a,b,d, Table 1). Nevertheless, the constrained mean values of the future North Atlantic $C_{ant*}$ uptake based on the optimal cuboids and the largest ellipsoid are consistently smaller than that of the original emergent constraint and offer a reduced uncertainty. Especially for the optimal cuboids, the regional optimisation leads to a narrowing down of our ensemble of well-performing models from five models (original emergent constraint) to three models (largest optimal cuboid) and finally down to two models (smallest optimal cuboid).

## 4 Discussion

With a multitude of model-projections available from several scenarios and model-generations, the desire to decrease the related model uncertainty based on a process-based understanding has increased. In this context, the concept of emergent constraint appears to be highly valuable and has become increasingly popular in recent years. Yet, the method has also attracted a lot of criticisms relating to, among others, the non-valid Gaussian assumption for the model ensemble, relationships between predictors and predictands that occur without any physical meaning behind them (Caldwell et al., 2014), non-robust emergent constraints that are not valid across different scenarios and model-ensembles, the assumption of linearity between predictor and predictand (Williamson and Sansom, 2019, who include a solution for testing the linearity assumption) and most prominently that the linear relationship of averaged values overly simplifies the complex interactions of many components and feedbacks (Schlund et al., 2020; Williamson and Sansom, 2019). Our study relates to the last point, but yet in a not-previously discussed manner: we advance the view that it is overly simplified to compare a regional average of the predictor (as often done in emergent constraints) to a regionally averaged observational value. We use regionally optimised emergent constraints to show that this course of action might deem a model to be 'fit' in the context of an emergent constraint but disregards that some aspects of the model's spatial distribution of the predictor within the considered region might be a misfit. Yet, the spatial distribution is of high importance for dynamical predictors that capture or rely on, for example, a transport from north to south. Here, the north-south distribution of the predictor is in fact an expression of its dynamical correctness. The spatial distribution is moreover especially important for predictands that are not evenly distributed within the considered region like the future North Atlantic $C_{ant*}$ uptake. This predictand has substantially higher $C_{ant*}$ uptake in higher latitudes such that a misfit in the north-south gradient of the winter $pCO_2^{sea}$-anomaly will have consequences for the correctness of the constrained value. While it can

be argued that a potentially easy approach to solve this problem is to additionally evaluate the spatial gradient of the predictor within the considered area (not done here), we note that not all parts of the considered region might be equally important for

the considered emergent constraint. Our regionally optimised emergent constraints point towards key areas for the emergent constraints (in terms of the predictor) and hence do reveal potential spatial mismatches only for highly important areas for the emergent constraint. Moreover, the identification of these key areas also allows us to uncover key dynamics behind the emergent constraint. We hence find our regionally optimised emergent constraints superior towards a simple gradient-analysis and recommend using it.

Regionally optimised emergent constraints can be applied to create new estimates of the predictand, which are potentially inconsistent with those of the original emergent constraint or with each other. In a review of emergent constraints, Williamson et al. (2021) noted that highly related predictors with different predictand-estimates indicate (i) persistent measurement biases and/or (ii) that the real world may not be sharing the same responses as the models and hence that a persistent error across the model ensemble exists. Our analysis does not consider the possibility of measurement biases as this is beyond the goal

of our study. Yet, we restrict measurement biases from playing a big role by assuming measurement errors generously. We hence use our regionally optimised emergent constraints to investigate potential inconsistencies within the model ensemble. In our case study, we note a potential model inconsistency for our first predictor, the winter $pCO_2^{sea}$-anomaly indicating that the south-north gradient of the model ensemble is not steep enough, i.e., the modelled northwards-propagation of related properties is too weak (see Section 3.2.2). However, the uncertainties around the observational estimates of the winter $pCO_2^{sea}$-anomaly

are large and do not allow us to be certain about this. We do not detect model inconsistencies for our second predictor, the fractional $C_{ant*}$-storage (see Section 3.3.2).

In our case study, both considered predictors are highly related to each other, and can therefore further be used to inform about inconsistencies between the simulated upper and interior ocean transport in terms of $C_{ant*}$. This is due to the fact that (i) the strength of the northward AMOC volume transport in the upper 500m drives the upper ocean properties in the high latitude

North Atlantic and hence the winter $pCO_2^{sea}$-anomaly (see Section 3.2.1); and concurrently, (ii) the strength of the southward AMOC volume transport in the interior ocean drives the effectiveness of surface-to-deep $C_{ant*}$-transport (see Section 3.3.1). Both parts are connected as the strength of the northward AMOC volume transport (i.e., the upper branch of the AMOC) is highly related to the strength of the southward AMOC volume transport (i.e., its lower branch). Specifically, the upper branch of the AMOC transports warm waters from the low latitude to the high latitude North Atlantic, thereby releasing heat to the

atmosphere (e.g., Rhein et al., 2011). Upon losing its heat, the water becomes denser and sinks. This densification links the warm, surface branch with the cold, deep return branch at regions of deep convection in the Nordic and Labrador Seas. For the Atlantic north of 26°N, volume conservation dictates that, for constant sea level, the net northward flow of upper waters balances the southward flow of deeper waters with a tolerance of 1Sv (McCarthy et al., 2015) such that there is a direct link between upper and lower branch of the AMOC, driving both our predictors and the predictand.

When dividing our newly constrained estimates into those associated with the upper ocean (0-500 m depth, i.e., those related to optimal rectangles and ellipses) and those of the deep ocean (below 500 m depth, i.e., those related to optimal cuboids), it can readily be seen that our observational constraints for the upper ocean predictors are systematically identifying models

with a higher future North Atlantic $C_{ant*}$ uptake to be better-performing than those for the deep ocean predictors (Figs. 5, 8a,b and Table 1). This is also reflected in our constrained mean values, which are 0.09-0.16 PgC/yr higher for the regionally optimised winter $pCO_2^{sea}$-anomaly than for the regionally optimised cuboids (see Table 1) and indicates a mismatch between the propagation from the upper ocean to the deep ocean for some of the models. We note that this is also confirmed for the optimal ellipsoids, where both smallest and medium-sized ellipsoids have a higher volume in the upper ocean (25% and 19% of their volumes are above 1000 m depth, respectively) and higher constrained mean values of the future North Atlantic $C_{ant*}$ uptake than the largest ellipsoid with only 6% of its volume above 1000 m. Based on our plausibility analysis for the optimal areas for the upper and deep ocean (Section 3.2.1 and 3.3.1), we found that the upper (0-500 m) North Atlantic northward volume transport at 30°N (2D-case) and the deep (700-4700 m) North Atlantic southward volume transport at 26°N (3D-case) are also plausible predictors for the future North Atlantic $C_{ant*}$ uptake and can be utilised to confirm this potential mismatch. Due to the limited observational availability, we only consider these volume transports at 26.5°N and for the time period 2005-2014. The resulting emergent constraints (see supplementary Figs. S1 and S7) confirm the assumed mismatch, identifying several models which are only well-performing for one of the volume-transport constraints. Only one model is able to perform well for all considered upper and deep ocean emergent constraints (CESM1-BGC, Figs. 5, 8, S1 and S7).

For both upper ocean and deep ocean constraints, the AMOC-observations come with lower observational uncertainty, yet the AMOC represents a purely physical constraint such that we consider the biogeochemical constraints as more closely related to the North Atlantic $C_{ant*}$ uptake and hence more plausible. This is reflected in the fact that they also offer very similar or higher correlations with the North Atlantic $C_{ant*}$ uptake when compared to the AMOC-constraints in the same ocean depth-range. A lower observational uncertainty in the biogeochemical constraints would hence be of high value.

## 5   Summary and conclusions

We applied a genetic algorithm to regionally optimize emergent relationships with the two-fold goal to (a) identify key model dynamics for the emergent constraint and model inconsistencies around them and (b) provide key areas where a narrow observational uncertainty is crucial for constraining future projections. We base the need for regional optimisation on the fact that emergent constraints are often related to dynamical features inherently coupled to spatial distributions. Hence, model performance of this dynamic can not be captured by one single averaged value as usually done for emergent constraints. As a case study to illustrate the usefulness of regional optimisation, we consider two previously established emergent constraints of the future carbon-uptake in the North Atlantic (Goris et al., 2018). The predictors of these emergent constraints are (i) the contemporary winter $pCO_2^{sea}$-anomaly, which is a surface quantity (2D-case) and (ii) the fraction of the North Atlantic $C_{ant*}$-storage, which is a surface-to-interior ocean quantity (3D-case). Both predictors relate to a model's ability to efficiently remove $C_{ant*}$ from the surface into the deep ocean.

The genetic algorithm was primarily adopted to find optimal regions for both predictors, such that cross-correlations between the regionally-optimized predictor-values and predictand-values are maximised. As emergent constraints are utilised to constrain the model spread, we only allowed the genetic algorithm to consider grid-points where the multi-model standard

deviation of the simulated predictors was larger than average. For the regional optimisation, we pre-defined a suite of different shapes and size-ranges, such that the genetic algorithm had to identify optimal ellipses and rectangles for the 2D-case and optimal ellipsoids and cuboids for the 3D-case with different sizes and volumes. Our consideration of different geometrical shapes and area sizes allows us to inspect in more detail where key regions for the model performance are, to determine if the simulated results for each of these regions are consistent with each other and to avoid spurious relationships.

Our results indicate that the genetic algorithm converges quickly for rectangles, ellipses and cuboids and slower for ellipsoids. After 100 iterations, the optimal solutions of the genetic algorithm provided higher cross-correlations than the original emergent constraints. The regional solutions of the 2D-case have cross-correlations between 0.848 and 0.863, that is 0.058 - 0.093 higher than that of the original emergent constraint of 0.79. The regional solutions of the 3D-case have cross-correlations between 0.966 and 0.987 and offered an improvement of 0.026 - 0.047 in comparison to that of the original emergent constraint of 0.94. The optimal predictor regions identify the Gulf Stream area at around 30°N to be central for our emergent constraint (2D-case) as well as the region of the interior ocean pathway of the southward volume transport (3D-case). Before following up on our two-fold goal of the regional optimisation, we investigated the plausibility of the newly identified optimal areas. The Gulf Stream is fundamental in transporting heat and nutrients to the north and is therefore key in determining a model's mixed layer depth as well as its productivity in high latitudes and hence its $C_{ant*}$-uptake. The interior ocean southward volume transport is fundamental for transporting $C_{ant*}$-saturated surface watermasses to the deep ocean and hence allowing for further high latitude $C_{ant*}$-uptake. These dynamical justifications led to the detection of two additional qualified predictors of the future North Atlantic $C_{ant*}$ uptake: the upper ocean northward volume transport between surface and 500 m depth at 30°N and the deep ocean southward volume transport between 700 m and 4700 m at 26°N. We note that the commonly used depth range of the northward maximum volume transport (surface to depth of maximum) did not allow for such high cross-correlations, neither at 26°N nor at 40°N. This indicates that the relation of maximum northward volume transport to the ocean carbon sink is not as robust as often assumed in modelling studies.

After this confirmation of the plausibility of the optimal areas, we used the regionally optimised emergent constraints to better understand the modelled dynamics of the predictors and potential inconsistencies around them. Though a typical emergent constraint should already have a solid physical background, its predictor is usually averaged over a large area such that the optimal areas found by the algorithm help to refine this knowledge. Our regional optimisation and the newly identified emergent constraints point us towards the fact that a correct simulation of the upper ocean and interior ocean volume transport is fundamental for a correct estimate of the future North Atlantic $C_{ant*}$ uptake. However, our results indicate that most models that perform well for the upper ocean volume transport do not perform well for the interior ocean volume transport and that most of the considered models do not capture the south-north gradient of the upper ocean northward volume transport well. It is questionable if a model that simulates the average upper ocean northward volume transport within observational constraints but not the related dynamical features like the south-north or vertical gradient of that transport can be considered a well-performing model. In future studies, we henceforth advise to combine the average values of the emergent constraint with a measure of spatial performance relating to the dynamical feature in question.

Though invaluable progress has been made through automated observational platforms like Argo (Argo, 2000) and analysis

tools like ESMValTool (Eyring et al., 2016b), observational networks and analyses of model projections are not growing in the same speed. Here, our regional optimisation of existing emergent constraints can be used to guide future monitoring strategies. We show this for the North Atlantic, where our results point towards the already employed RAPID array and prove that the genetic algorithm is able to provide meaningful results. We note, however, that the RAPID array takes purely physical observations, though our localised emergent constraints show that additional measures of carbon-storage would fundamentally improve our understanding of the $C_{ant*}$-uptake in the North Atlantic.

To our knowledge, this is the first time that a regional optimisation of emergent constraints has been carried out. The results are of high value as the use of emergent constraints in the realm of climate projections has gained a lot of momentum in the last decade (see Williamson et al., 2021, for a review of existing emergent constraints for climate sensitivities) due to the fast growing number of models taking part in coordinated model exercises associated with future projections (e.g. Balaji et al., 2018). Here, a regional optimisation can be valuable to identify model inconsistencies in terms of spatial gradients and at the same time point towards areas where a reduction of observational uncertainties is most useful.

*Code and data availability.* The code of the genetic algorithm including the relevant input and output data is available through Johannsen (2022a, https://doi.org/10.5281/zenodo.7037947) and Johannsen (2022b, https://doi.org/10.5281/zenodo.7037981) for our North Atlantic 2D and 3D case study, respectively.

## Appendix A: Observational Estimates

For observational estimates of the contemporary winter $pCO_2^{sea}$-anomaly (depicted in Fig. 1b and Fig. 5), we utilised a neural-network-based interpolated $pCO_2^{sea}$ product (Landschützer et al., 2017, https://doi.org/10.7289/v5z899n6, version 2.2). Specifically, we calculated the contemporary winter $pCO_2^{sea}$-anomaly as a decadal average based on the 'spco2_smoothed'-variable for the years 1990-1999. We note that it would have been possible to focus on other time-frames. However, we decided to consider the same time-frames as in Goris et al. (2018), so that an easy comparison of previous and new results is possible. As the utilised $pCO_2^{sea}$ database does not include an error-estimate, we utilise the error-estimate of the supplementary information of Landschützer et al. (2018), where the neural-network product is analysed for seasonal mean biases for 4 broad latitudinal bands. Results for summer and winter biases of the data-product for the latitudinal bands of 10-40°N and 40-65°N show that the biases are randomly spread around 0, but do show substantial variability. We apply the largest detected seasonal bias of these latitudinal bands of about $\pm 14$ $\mu$atm as our uncertainty range of the observational estimate of the contemporary winter $pCO_2^{sea}$-anomaly.

For observational estimates of the contemporary fractional $C_{ant*}$-storage (depicted in Fig. 1d and Fig. 8), a mapped climatology of anthropogenic carbon ($C_{ant}$) has been used (Lauvset et al., 2016, https://doi.org/10.7289/v5kw5d97, version GLODAPv2.2016, mapped). We note that there is a difference between this data-product and our modelled estimates as the data product describes $C_{ant}$ and the modelled estimates describes $C_{ant*}$, i. e., a combination of the anthropogenic component

of the carbon cycle combined with climate change–induced differences. Yet, for the time span of the historical simulation, the climate-change induced differences are small and it is possible to use $C_{ant}$ as an approximation of $C_{ant*}$ (Frölicher et al., 2015). The observation-based data-product of the $C_{ant}$-storage is normalized to the year 2002. We therefore compare it to the simulated fractional $C_{ant*}$-storage in the time-frame 1997-2007, abbreviated as 1997s. For conversion of the data product from $\mu$mol/kg to PgC, we utilised a mean ocean standard density of 1036 kg/m$^3$ (Pawlowicz, 2013). Furthermore, we linearly interpolated the data-product onto the considered depth-levels. Though the data-product includes estimates of a mapping error, a comprehensive error estimate containing observational, methodological and mapping error is not available. In lack of such an estimate, we follow the approach of Goris et al. (2018) and use an error estimate of $\pm10\%$ for the observational estimate of the fractional $C_{ant}$-storage below 1000 m accumulated over the whole North Atlantic (Fig. 1d). In order to get an error estimate for the fractional $C_{ant}$-storage within our optimal cuboids and ellipsoids, we utilise the error-estimate of $\pm29\%$ for the $C_{ant}$-storage of the North Atlantic (Steinfeldt et al., 2009). The simple assumption of an error of 29% for every grid point leads to the same factor in numerator and denominator and results in an error estimate of zero for the fractional $C_{ant}$-storage. Only a spatially heterogeneously distributed error of the $C_{ant}$-storage leads to a non-zero error estimate for the fractional $C_{ant}$-storage. As such an error estimate is missing, we simply assume an error of $\pm$ 29% within our optimal areas but assume no error for other grid points of the North Atlantic, which are taken into account to build the fractional measure. When assuming an error of $\pm29\%$ for the $C_{ant}$-values within our largest optimal cuboid, we obtain an error of +3.78% and -4.12% for the fractional $C_{ant}$-storage. In order to obtain an evenly distributed error around the mean value, we chose the error-value larger in absolute values, yielding an error of $\pm4.12\%$ for the largest cuboid. We follow the same procedure for the other optimal volumes.

For observational estimates of the contemporary strength of northward and southward volume transport (depicted in Figs. S1 and S7), data from the RAPID-Meridional Overturning Circulation and Heatflux Array-Western Boundary Time Series array at 26°N have been employed (Frajka-Williams et al., 2021, https://doi.org/10/gwqg). RAPID-observations are only available from April 2004 onward, though our application of the genetic algorithm considers the years 1990-1999 (2D-case) and 1997-2007 (3D-case) for our regional optimisation. Due to lacking observations in the time-frames of interest, we build a decadal average of the observations of the AMOC streamfunction profile for the years 2005-2014. When needing to access accumulations of the streamfunction over differing depth ranges with boundaries at surface, 500 m, 700 m and 4700 m, we utilise the observed depths that are closest to these boundaries, i.e. surface, 496 m, 694 m and 4696 m. We consider these depth-values to be close enough to the desired boundaries such that no interpolation is necessary. Annual error estimates between 0.9 and 1.3 Sv are given for maximum northward transport estimates of the years 2004 to 2014 (https://rapid.ac.uk/rapidmoc/rapid_data/README_ERROR.pdf, accessed in November 2021). We employ the estimate of 1.3Sv as our observational error estimate. We note, however, that we do not utilise the maximum northward volume transport estimate directly but accumulate differing depths of the streamfunction profile. This might lead to the error-estimate being imprecise.

## Appendix B:  Streamfunction values for our CMIP5 ensemble

Several of the here considered models did not provide the Atlantic Meridional Overturning Circulation to the CMIP5 database. Therefore, we utilised AMOC streamfunction values calculated with monthly mean meridional currents as described in Mecking et al. (2017).

*Author contributions.*  NG, KJ and JT designed the application of the genetic algorithm. KJ implemented the genetic algorithm. NG prepared the ocean biogeochemical input data, extended the analysis to include the transport, illustrated the results and established and analysed the
new emergent constraints. NG wrote the manuscript with assistance of KJ and JT.

*Competing interests.*  The authors declare that they have no conflict of interest.

*Acknowledgements.*  We would like to thank Jennifer Mecking for providing us with calculated AMOC streamfunction values for our model ensemble as well as Timothée Bourgeois for making his MATLAB-function for emergent constraints available for our research. We would also like to thank three reviewers for their constructive comments. This work was supported by the Norwegian Research Council through the
project COLUMBIA and CE2COAST (Grant 275268 and 318477). High-performance computing and storage resources were provided by the Norwegian Research Infrastructure Services (projects nn1002k and ns1002k). We acknowledge the World Climate Research Programme's Working Group on Coupled Modelling, which is responsible for CMIP, and we thank the climate modeling groups for producing and making available their model output. For CMIP, the U.S. Department of Energy's Program for Climate Model Diagnosis and Intercomparison provides coordinating support and led development of software infrastructure in partnership with the Global Organization for Earth System
Science Portals. Data from the RAPID AMOC monitoring project is funded by the Natural Environment Research Council and are freely available from www.rapid.ac.uk/rapidmoc.

## Emergent constraints of the North Atlantic future $C_{ant*}$-uptake

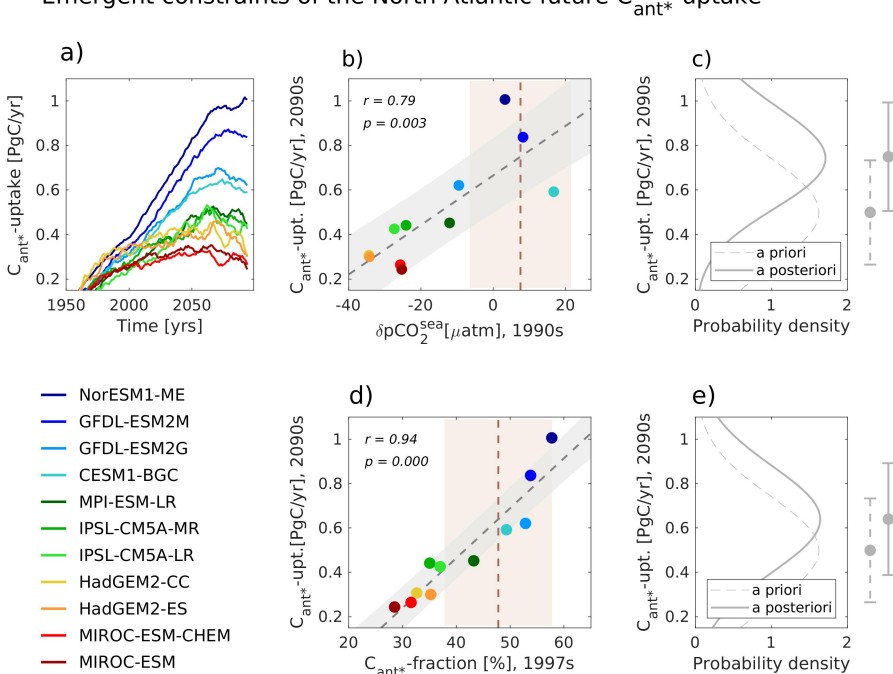

**Figure 1.** Illustration of two emergent constraints of the future North Atlantic $C_{ant*}$-uptake, both considering the same ensemble of 11 CMIP5-models under a high $CO_2$ future. (a) Temporal evolution (10-year running averages) for the North Atlantic $C_{ant*}$-uptake (predictand). Projected North Atlantic $C_{ant*}$-uptake for the years 2090-2099 against (b) the mid-to-high latitude winter $pCO_2^{sea}$-anomaly, 1990-1999 (predictor 1) and d) the fraction of the North Atlantic $C_{ant*}$ stored below 1000 m depth, 1997-2007 (predictor 2). (b,c) Scatter-plots of model results (color coding of models indicated in legend), best fit linear regression (grey dashed lines) including the interval of the 68% projection uncertainty (grey shading) as well as observational constraints and their uncertainties (brown dashed lines and light brown shading). (c,e) Prior- and after-constraint probability density functions and their associated new estimates of the future North Atlantic $C_{ant*}$-uptake for the years 2090-2099 (on the right side of the panels). See Appendix A for a detailed description of the considered observational estimates.

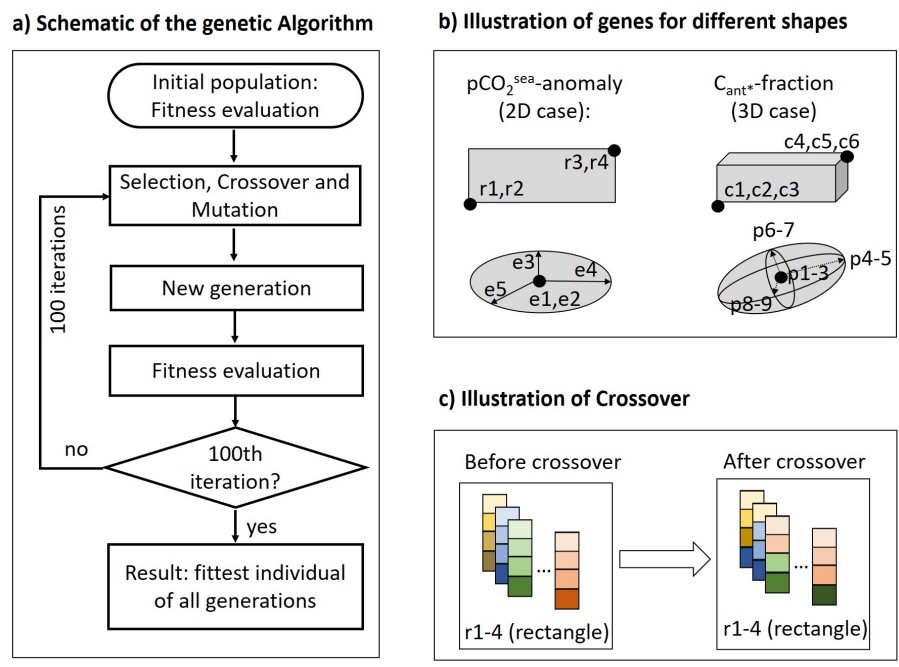

**Figure 2.** Schematic illustration for the experimental set-up of our application of the genetic algorithm. The panels illustrate a) one iteration of the algorithm, b) genes chosen to represent rectangles, ellipses, cuboids and ellipsoids as well as c) visualisation of a crossover for a population of rectangles.

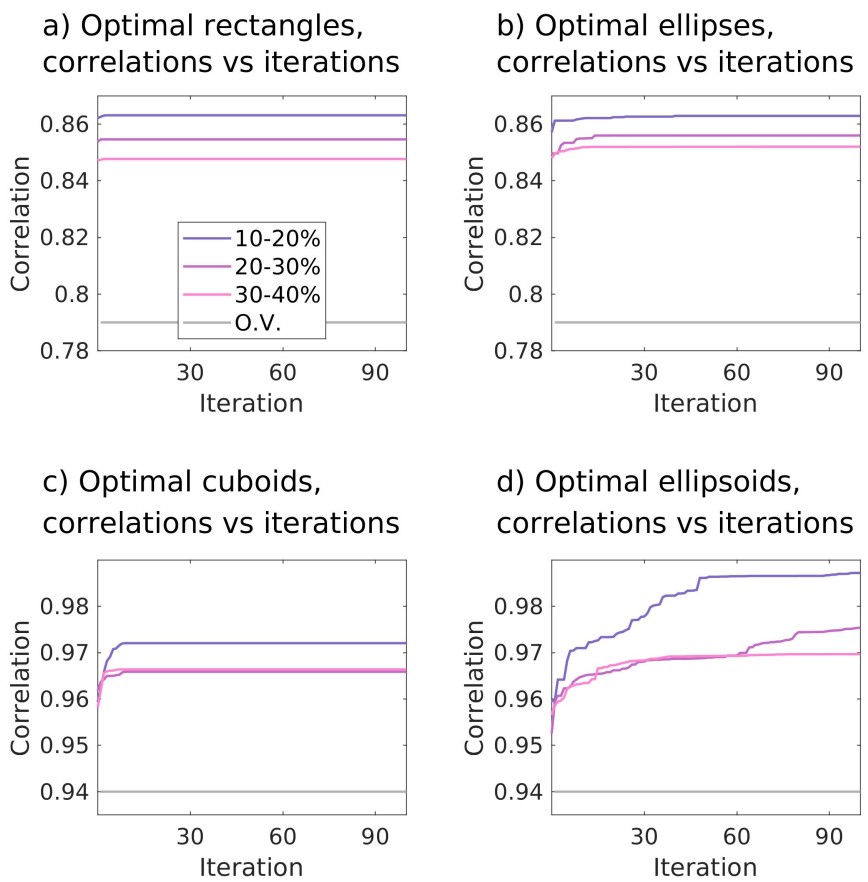

**Figure 3.** Iteration (population) versus cross-correlations for our application of the genetic algorithm. The correlation coefficients are calculated between contemporary values of the predictor within the regions identified by the genetic algorithm and future values of the predictand. Illustrated are the individuals with the highest cross-correlation of each population (i.e., per iteration). The color-coding of the lines point towards the area and volume sizes that characterise the identified regions. The gray lines illustrate the cross-correlation without regional optimisation of the predictor.

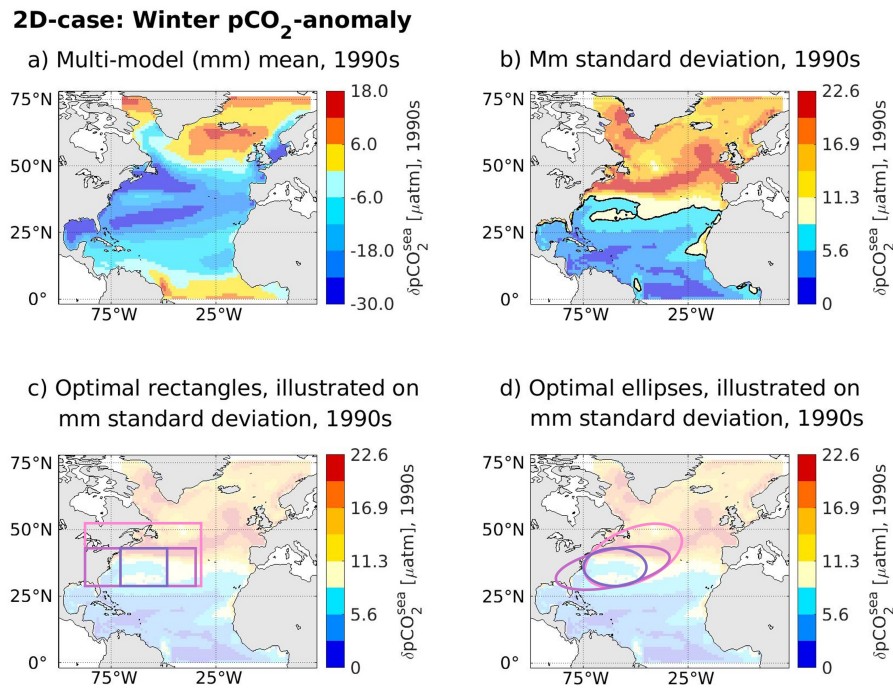

**Figure 4.** Contemporary winter $pCO_2^{sea}$-anomaly and associated optimal regions as identified by the genetic algorithm. For the contemporary winter $pCO_2^{sea}$-anomaly of our considered model-ensemble, panel (a) illustrates the multi-model mean, while panel (b) displays the multi-model standard deviation. Panels (c,d) display the optimal regions identified by the genetic algorithm on top of the multi-model standard deviation (here with an added transparency of 70%) with non-eligible points colored in different shades of blue (separated with a black contour line in panel (b)). Optimal regions are visualised according to shapes, with panel (c) visualising rectangles and panel (d) visualising ellipses. The color-coding of the lines indicate different area-conditions that were imposed on the optimal areas (dark lilac lines: area-size of 10-20%, light lilac lines: area-size of 20-30% and pink lines: area-size of 30-40% of the surface of the North Atlantic).

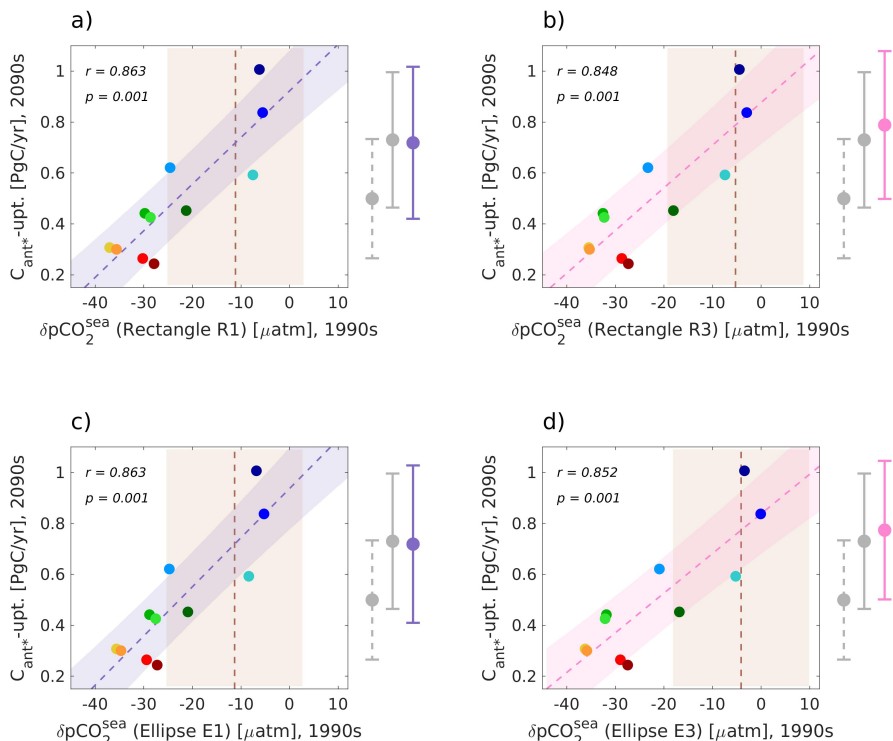

**Figure 5.** Illustration of emergent constraints between different realisations of the regionally optimised winter $pCO_2^{sea}$-anomaly (predictor) for the years 1990-1999 and the future North Atlantic $C_{ant*}$-uptake (predictand) for the years 2090-2099 for our model ensemble. Emergent constraints for optimal regions of different area-size conditions in shape of rectangles are visualised in the top panels (R1: 10-20%, R3: 30-40% of the considered area), while those in shape of ellipses are visualised in the bottom panels (E1: 10-20%, E3: 30-40% of the considered area). All panels show scatter-plots (color coding of models as in Figure 1), best fit linear regression (R1/E1: lilac line, R3/E3: pink line) including the interval of the 68% projection uncertainty (R1/E1: lilac shading, R3/E3: pink shading), cross-correlations between simulated predictor and predictand as well as mean observational constraints and their uncertainties (brown dashed lines and light brown shading). Associated estimate for the unconstrained model ensemble (grey dashed bars), the original emergent constraint (grey bars) and the regionally optimised emergent constraint (lilac/pink bars) are shown on the right side of the panels. See Appendix A for a detailed description of the considered observational estimates.

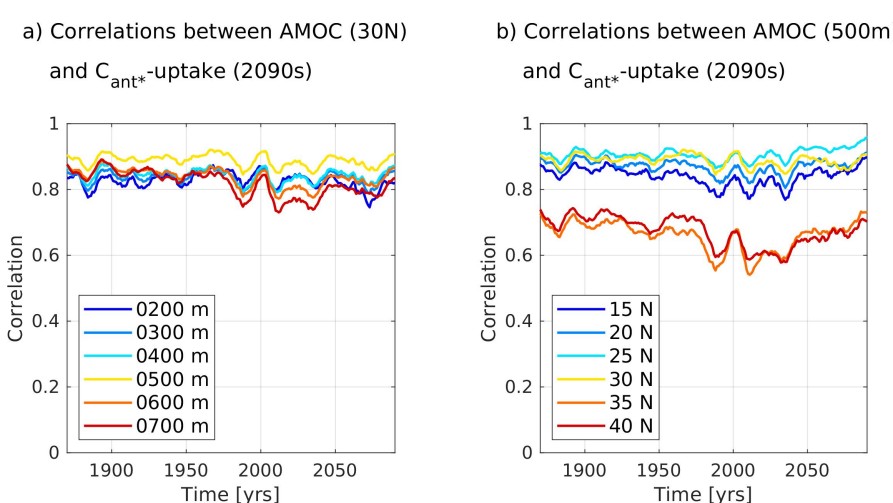

**Figure 6.** Time series of cross-correlations between 10-year running averages of the simulated upper branch of the Atlantic Meridional Overturning Circulation (AMOC) and the future North Atlantic $C_{ant*}$-uptake (2090s) for our model ensemble. The upper branch of the AMOC is expressed as accumulated northward volume transport between surface and a lower depth boundary at a certain latitude. Panel (a) shows results for $30°$N and a varying lower depth boundary, while panel (b) shows results for a lower depth boundary of 500 m and varying latitudes.

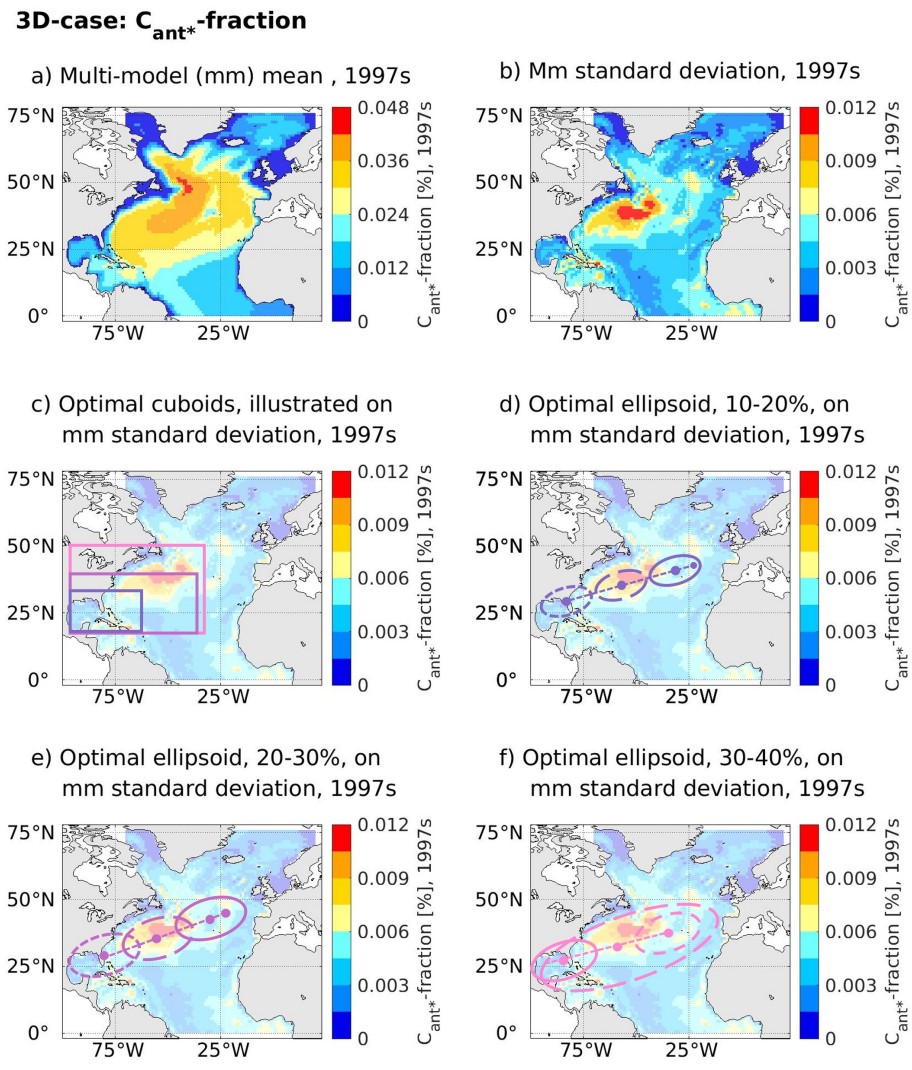

**Figure 7.** Contemporary fraction of the North Atlantic $C_{ant*}$ and associated optimal regions as identified by the genetic algorithm. For the depth-integrated contemporary fraction of the North Atlantic $C_{ant*}$ of our considered model-ensemble, panel (a) illustrates the multi-model mean, while panel (b) displays the multi-model standard deviation. Panels (c,d,e,f) display the optimal regions identified by the genetic algorithm on top of the multi-model standard deviation (here with an added transparency of 70%). Optimal regions are visualised according to shapes, with panel (c) visualising cuboids with volume-sizes of 10-20% (dark lilac lines), 20-30% (light lilac lines) and 30-40% (pink lines) of the North Atlantic. Panels (d,e,f) visualise ellipsoids of different volumes via illustration of their mid-points (dots) and outlines for the depth-planes 500-660 m (continuous line), 2500-2600 m (long dashed line) and 4500-4600 m (dashed line) and their depth-following principal axis (line connecting the mid-points). In panels (d,e), the midpoint of the surface plane is additionally illustrated.

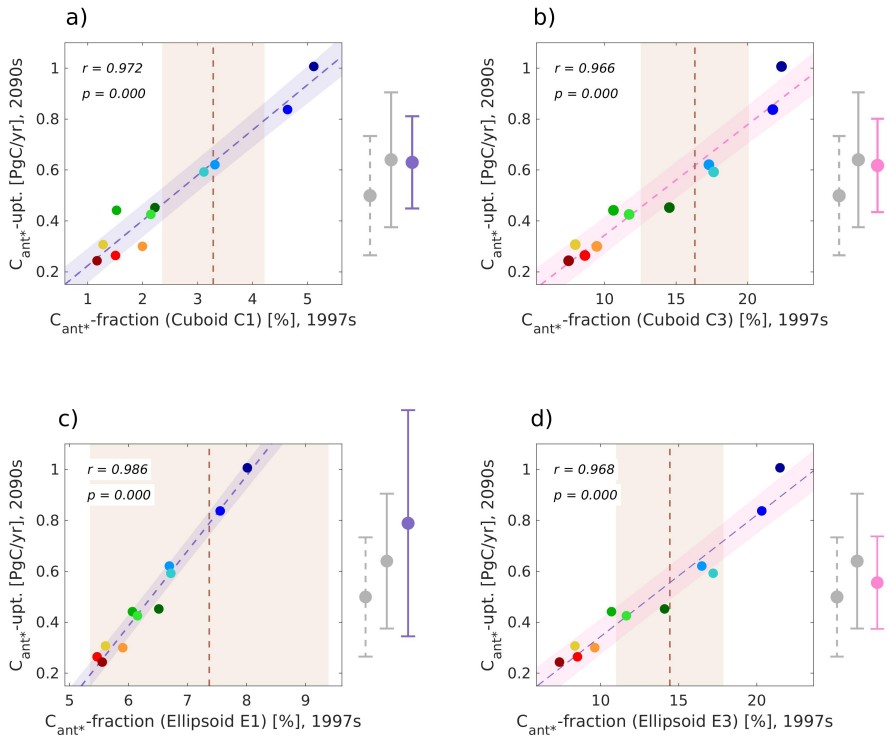

**Figure 8.** Illustration of emergent constraints between different realisations of the regionally optimised $C_{ant*}$-fraction (predictor) for the years 1997-2007 and the future North Atlantic $C_{ant*}$-uptake (predictand) for the years 2090-2099 for our model ensemble. Emergent constraints for optimal regions of different volume-size conditions in shape of cuboids are visualised in the top panels (C1: 10-20%, C3: 30-40% of the considered volume), while those in shape of ellipsoids are visualised in the bottom panels (E1: 10-20%, E3: 30-40% of the considered volume). All panels show scatter-plots (color coding of models as in Figure 1), best fit linear regression (R1/E1: lilac line, R3/E3: pink line) including the interval of the 68% projection uncertainty (R1/E1: lilac shading, R3/E3: pink shading), cross-correlations between simulated predictor and predictand as well as mean observational constraints and their uncertainties (brown dashed lines and light brown shading). Associated estimate for the unconstrained model ensemble (grey dashed bars), the original emergent constraint (grey bars) and the regionally optimised emergent constraint (lilac/pink bars) are shown on the right side of the panels. See Appendix A for a detailed description of the considered observational estimates.

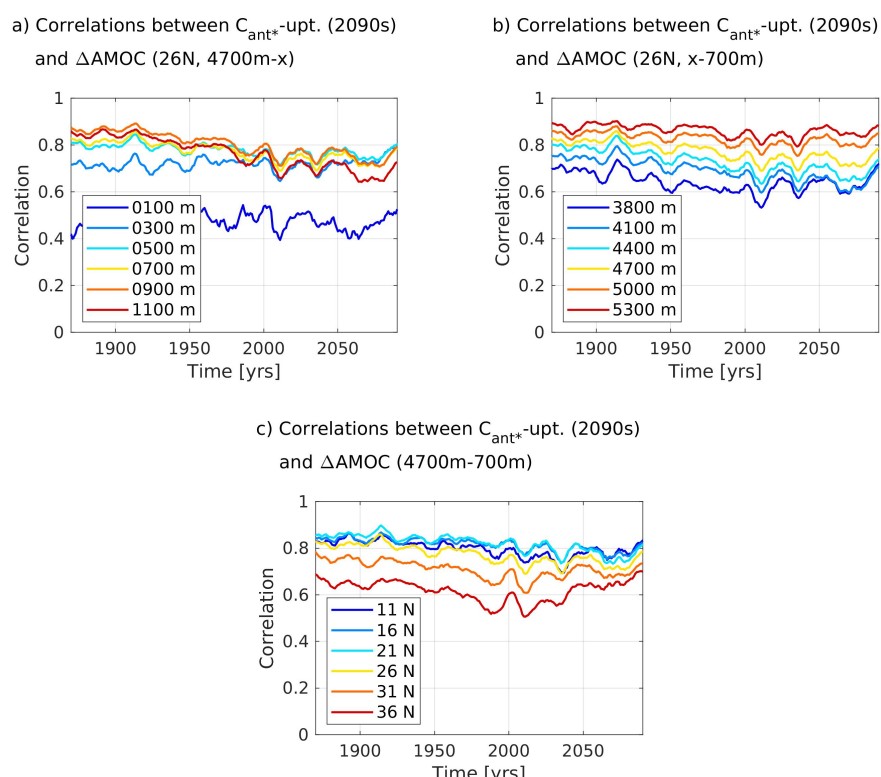

**Figure 9.** Time series of cross-correlations between 10-year running averages of the simulated lower branch of the Atlantic Meridional Overturning Circulation (AMOC) and the future North Atlantic $C_{ant*}$-uptake (2090s) for our model ensemble. The lower branch of the AMOC is expressed as accumulated southward volume transport between a higher depth boundary and a lower depth boundary at a certain latitude. Panel (a) shows results for 26°N, a lower depth boundary at 4700 m and a varying higher depth boundary, while panel (b) shows results for 26°N, a higher depth boundary at 700 m and a varying lower depth boundary and panel (c) for a higher depth boundary at 700 m, a lower depth boundary at 4700 m and varying latitudes.

**Table 1.** Constrained estimates of the future North Atlantic $C_{ant*}$-uptake based on regionally optimised predictors. Listed are the predictors, their realms (upper ocean: above 500 m, deep ocean: below 500 m depth) and considered time-frames as well as the associated constrained estimates of the future North Atlantic $C_{ant*}$-uptake. Different size-ranges of the optimal areas are indicated with numbers (1: area-size of 10-20%, 2: area-size of 20-30% and 3: area-size of 30-40% of the considered area).

| predictor | realm | time-frame | constrained $C_{ant*}$-upt. |
|---|---|---|---|
| $\delta pCO_2^{sea}$ (Ellipse E1) | upper ocean | 1990-1999 | $0.72 \pm 0.31$ PgC/yr |
| $\delta pCO_2^{sea}$ (Ellipse E2) | upper ocean | 1990-1999 | $0.72 \pm 0.28$ PgC/yr |
| $\delta pCO_2^{sea}$ (Ellipse E3) | upper ocean | 1990-1999 | $0.77 \pm 0.27$ PgC/yr |
| $\delta pCO_2^{sea}$ (Rectangle R1) | upper ocean | 1990-1999 | $0.72 \pm 0.30$ PgC/yr |
| $\delta pCO_2^{sea}$ (Rectangle R2) | upper ocean | 1990-1999 | $0.73 \pm 0.31$ PgC/yr |
| $\delta pCO_2^{sea}$ (Rectangle R3) | upper ocean | 1990-1999 | $0.79 \pm 0.29$ PgC/yr |
| $\Delta$AMOC, 26°N (0-500 m) | upper ocean | 2005-2014 | $0.74 \pm 0.18$ PgC/yr |
| $C_{ant*}$-fraction (Ellipsoid E1) | water column | 1997-2007 | $0.79 \pm 0.44$ PgC/yr |
| $C_{ant*}$-fraction (Ellipsoid E2) | water column | 1997-2007 | $0.73 \pm 0.36$ PgC/yr |
| $C_{ant*}$-fraction (Ellipsoid E3) | water column | 1997-2007 | $0.55 \pm 0.18$ PgC/yr |
| $C_{ant*}$-fraction (Cuboid C1) | deep ocean | 1997-2007 | $0.63 \pm 0.18$ PgC/yr |
| $C_{ant*}$-fraction (Cuboid C2) | deep ocean | 1997-2007 | $0.62 \pm 0.17$ PgC/yr |
| $C_{ant*}$-fraction (Cuboid C3) | deep ocean | 1997-2007 | $0.62 \pm 0.18$ PgC/yr |
| $\Delta$AMOC, 26°N (700-4700 m) | deep ocean | 2005-2014 | $0.57 \pm 0.20$ PgC/yr |

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
