# Peer review of "The emergence of Gulf Stream and interior western boundary as key regions to constrain the future North Atlantic Carbon Uptake"

_Geoscientific Model Development, 2022_

## Author Comment (AC3)

**Answer to Review #1, https://doi.org/10.5194/gmd-2022-152-RC1**

**We thank the reviewers for the thoughtful review. Detailed point by point responses to the major and minor comments are given below, with reviewers' comments in black and answers of the authors in blue.**

Goris et al., Gulf Stream and interior western boundary volume transport as key regions to constrain the future North Atlantic Carbon Uptake

This study aimed for regional optimization of the emergent constraints for projecting future North Atlantic carbon uptake. A previous study (Goris et al., 2018) identified two indicators, i.e., seasonal  $pCO_2^{sea}$  anomaly in middle-to-high latitude and fraction of anthropogenic carbon inventory below 1000m, for future carbon uptake projection in the North Atlantic. The authors apply a genetic algorithm to further find out which spatial area and depth ranges are crucial for emergent relationships. This study is scientifically interesting to constrain the projections of the North Atlantic Ocean carbon uptake, and also practically provide guidance for monitoring and observational strategies. However, this manuscript needs some further work and clarification to be published.

**We thank the reviewer for the encouraging and constructive comments.**

**Major comments:**

-Inconsistency of the season for  $pCO_2^{sea}$  anomaly: it is winter time  $pCO_2^{sea}$  in this study, but the cited paper (Goris et al., 2018) used summer time  $pCO_2^{sea}$ . The correlations should be reversed but are the same in both manuscripts.

**Answer 1:** In Goris et al. (2018), the negative mean summer  $pCO_2^{sea}$ -anomaly is utilised in parts of the manuscript so that positive correlations can be visualised (correlations with the summer time  $pCO_2^{sea}$ -anomaly are negative). As Goris et al. (2018) define the mean summer  $pCO_2^{sea}$ -anomaly to be the averaged May–October  $pCO_2^{sea}$ -value minus the annual  $pCO_2^{sea}$ -value of the same year, the negative mean summer  $pCO_2^{sea}$ -anomaly equals the mean winter  $pCO_2^{sea}$ -anomaly (November-April). We regret not having expanded on this in the original manuscript and have added an explanation about the relation between mean winter  $pCO_2^{sea}$ -anomaly and negative mean summer  $pCO_2^{sea}$ -anomaly as used in Goris et al. (2018) to our revised manuscript.

-The constrained relationship of winter  $pCO_2^{sea}$  anomaly is relative small (r=0.79), maybe it is because the definition of winter months (November to April) in this study. The variations in different months should be quite different, especially in the transit seasons, i.e., spring and autumn. Definition of the focus season with less months, e.g. December to February, or January to March, might end up with clear relationship and higher correlation.

**Answer 2:** We thank the reviewer for this suggestion. It is the goal of our manuscript to show how a genetic algorithm can be utilised to regionally constrain already existing emergent constraints. As a showcase, we use the already existing emergent constraints from Goris et al. (2018). Here, it is not our goal to redefine these existing emergent constraints and we note that even correlations weaker than r=0.79 are commonly applied in the context of emergent constraints (e.g., Qu et al. 2018, Selten et al., 2020, Mystakidis et al., 2017, Tokarska et al., 2020). We have added a similar explanation to our revised manuscript.

We would like to expand that we did check other definitions of seasonal pCO2sea anomalies when preparing the manuscript of Goris et al. (2018) and found that less (or a different selection of) months do not provide higher correlations. This is because the seasonal pCO2sea anomaly is chosen as a measure to capture the difference between models whose pCO2sea seasonality is driven by variations in dissolved inorganic carbon (driven by mixed layer depth and biological production) and models whose pCO2sea seasonality is driven by variations in sea surface temperature. As the here considered models have different timings for their peak in biological production (ranging from May to July) and as seasonal warming and biological production is not in phase (the peak in seasonal warming occurs in August for the here considered models), it is necessary to at least cover the months from May to August to capture the different seasonal drivers at play. However, further investigation revealed that seasonal warming is a dominant driver until the month of October. We have added a similar explanation in our revised manuscript.

-This study presented several predictors including the two from Goris et al. (2018). As shown in Fig. 8, each predictor provides a different estimate of the constrained range of future North Atlantic carbon uptake. Which estimate is more plausible?

Answer 3: We thank the reviewer for this very interesting question. We would like to first point out that our preprint states that "all newly constrained values for the future North Atlantic Cant+ uptake are consistent which each other, i.e. the uncertainties around the constrained mean values are large enough for the solutions to not contradict each other" (lines 410ff). Accordingly, all estimates can be true at the same time. When it comes to the mechanisms that we are using to constrain the future North Atlantic Cant\* uptake, it is our understanding that both emerging mechanisms are not completely independent in determining the Cant\*-uptake strength in the North Atlantic and they should not be viewed as two separate constraints. That is (i) the AMOC strength in the upper 500m drives the cold waters and the productivity levels in the high latitude North Atlantic; concurrently, (ii) the strength of the lower limb AMOC, which relates to the strength of upper limb AMOC, drives the effectiveness of surface-to-deep Cant\* transport. For both upper ocean and deep ocean constraints, the AMOC-observations come with lower observational uncertainty, yet they represent a purely physical constraint such that we consider the biogeochemical constraints as more closely related to the North Atlantic Cant\* uptake and hence more plausible. This is reflected in the fact that they also offer higher correlations with the North Atlantic Cant\* uptake when compared to the AMOC-constraints in the same ocean depth-range. A lower observational uncertainty in the biogeochemical constraints would hence be of high value. We have added a similar explanation to our revised manuscript.

-How are the uncertainty range of the predictand Cant\*-uptake in Fig. 1, 8 and Table 1 calculated? I suppose they should be determined by the cross points of the linear regression line and the vertical lines of the observational uncertainty, but apparently it is not the case as shown in Fig. 1(c and e) and Fig. 8.

**Answer 4:** Indeed, this is not the case. We had noted in lines 401ff that "For details of the method that we utilise to calculate the unconstrained and observationally constrained estimates of the future North Atlantic Cant\* uptake, the reader is referred to Bourgeois et al. (2022)." However, we understand that this statement is appearing too late in the manuscript and that a short introduction of the method would be helpful to the reader. Our method of estimating the constrained estimate follows the original approach of Cox et al. (2013). Here, the unconstrained estimate is given by the model mean and its uncertainty by the multi-model standard deviation. Assuming that all models are equally likely to simulate the true future

North Atlantic Cant\* uptake and are sampled from a Gaussian distribution, a probability density function (PDF) can be calculated for the unconstrained estimate using model mean and standard deviation. Similarly, a PDF of the observational estimate and of the linear regression between predictor and predicant is established. For the observationally constrained future North Atlantic Cant\* uptake, a conditional PDF is calculated by integrating over the product of the PDF of the observational estimate and the PDF of the linear regression. The observationally constrained estimate equals the expected value of the conditional PDF and the uncertainty of the estimate is given by its standard deviation. We have added a similar short introduction of the method to our revised manuscript, and, within the revised manuscript, we refer to this short introduction in the captions of Figures 1 and 6 as well as Table 1.

- It is not very clear how to interpret Fig. 5 and Fig. 7. The results in the two figures seem to contradict each other, both upper ocean (Fig. 5) and deeper ocean (Fig. 7) have high correlations. How to combine the information? In addition, L467-469: "...the deep ocean southward volume transport between 700m-4700m at 26N." Is this statement based on Fig. 7? This figure shows the 700m-5300m and 21N has reached the largest correlations.

**Answer 5:** We refer to Answer 3 and add that the strength of the northward mass transport with the AMOC (i.e., its upper cell or upper limb) is highly related to the strength of the southward mass transport with the AMOC (i.e., its lower cell or lower limb). Specifically, the upper branch of the AMOC transports warm waters from the low latitude to the high latitude North Atlantic, thereby releasing heat to the atmosphere (e.g., Rhein et al., 2011). Upon losing its heat, the water becomes denser and sinks. This densification links the warm, surface limb with the cold, deep return limb at regions of deep convection in the Nordic and Labrador Seas. For the Atlantic north of 26°N, volume conservation dictates that, for constant sea level, the net northward flow of upper waters balances the southward flow of deeper waters with a tolerance of 1Sv (McCarthy et al., 2015) such that there is a direct link between upper and lower cell of the AMOC. Though we are not considering the exact boundaries of the upper and lower limb, we have made an additional Figure (Figure R1, below) showing that the tight connection between upper and lower limb also holds for the here considered depth ranges. We have added a similar explanation to our revised manuscript.

The statement in L467-469 is based on two considerations: (1) While Figure 7 indicates that the southward transport between 700m-5300m reaches the highest correlations with the future North Atlantic  $C_{ant*}$  uptake, the amount of  $C_{ant*}$  that can be transported below 4700m is negligible. (2) We considered 26N instead of 21N as an observational constraint is available at 26N, while this is not the case at 21N. While we had denoted both considerations at different places in the preprint, we failed to summarise this more prominently. We have added a more prominent summary in our revised manuscript.

**Figure R1:** Scatter plot of simulated mass transport at 26N in the upper 500m versus simulated mass transport at 26N between 700 and 4700m. The associated correlation coefficient is indicated in the upper left, the considered models and their color-coding is denoted in the legend.

**Minor comments:**

-The information of figures are incomplete. I would suggest the authors to ensure all the figures are more or less self-explainable.

**Answer 6:** We thank the reviewer for pointing this out. As answer to this comment, we have re-done Figure 3 (see Answer 7), added units and expanded the captions for Figures 4 and 6 (see Answer 8) and presented the models in colours in Figure 8 (see Answer 9).

Fig. 3: the titles of x-axis and y-axis are missing, the y-axis' title is relative easy to guess, but the x-axis is not so straightforward. The readers need to check back and forth of the context to figure out that it should be number of iterations.

**Answer 7:** We apologize for overlooking this and thank the reviewer for pointing this out. Considering the previous comment of the reviewer, we decided to re-do Fig. 3 (depicted below) so that it hopefully is now self-explanatory and easier to understand.

---

## Author Comment (AC4)

Answer to Review #1, https://doi.org/10.5194/gmd-2022-152-RC1

**We thank the reviewers for the thoughtful review. Detailed point by point responses to the major and minor comments are given below, with reviewers´ comments in black and answers of the authors in blue.**

Goris et al., Gulf Stream and interior western boundary volume transport as key regions to constrain the future North Atlantic Carbon Uptake

This study aimed for regional optimization of the emergent constraints for projecting future North Atlantic carbon uptake. A previous study (Goris et al., 2018) identified two indicators, i.e., seasonal $pCO_2^{sea}$ anomaly in middle-to-high latitude and fraction of anthropogenic carbon inventory below 1000m, for future carbon uptake projection in the North Atlantic. The authors apply a genetic algorithm to further find out which spatial area and depth ranges are crucial for emergent relationships. This study is scientifically interesting to constrain the projections of the North Atlantic Ocean carbon uptake, and also practically provide guidance for monitoring and observational strategies. However, this manuscript needs some further work and clarification to be published.

We thank the reviewer for the encouraging and constructive comments.

**Major comments:**

-Inconsistency of the season for $pCO_2^{sea}$ anomaly: it is winter time $pCO_2^{sea}$ in this study, but the cited paper (Goris et al., 2018) used summer time $pCO_2^{sea}$. The correlations should be reversed but are the same in both manuscripts.

**Answer 1:** In Goris et al. (2018), the negative mean summer $pCO_2^{sea}$-anomaly is utilised in parts of the manuscript so that positive correlations can be visualised (correlations with the summer time $pCO_2^{sea}$-anomaly are negative). As Goris et al. (2018) define the mean summer $pCO_2^{sea}$-anomaly to be the averaged May–October $pCO_2^{sea}$-value minus the annual $pCO_2^{sea}$-value of the same year, the negative mean summer $pCO_2^{sea}$-anomaly equals the mean winter $pCO_2^{sea}$-anomaly (November-April). We regret not having expanded on this in the original manuscript and have added an explanation about the relation between mean winter $pCO_2^{sea}$-anomaly and negative mean summer $pCO_2^{sea}$-anomaly as used in Goris et al. (2018) to our revised manuscript.

-The constrained relationship of winter $pCO_2^{sea}$ anomaly is relative small (r=0.79), maybe it is because the definition of winter months (November to April) in this study. The variations in different months should be quite different, especially in the transit seasons, i.e., spring and autumn. Definition of the focus season with less months, e.g. December to February, or January to March, might end up with clear relationship and higher correlation.

**Answer 2:** We thank the reviewer for this suggestion. It is the goal of our manuscript to show how a genetic algorithm can be utilised to regionally constrain already existing emergent constraints. As a showcase, we use the already existing emergent constraints from Goris et al. (2018). Here, it is not our goal to redefine these existing emergent constraints and we note that even correlations weaker than r=0.79 are commonly applied in the context of emergent constraints (e.g., Qu et al. 2018, Selten et al., 2020, Mystakidis et al., 2017, Tokarska et al., 2020). We have added a similar explanation to our revised manuscript.

We would like to expand that we did check other definitions of seasonal $pCO_2^{sea}$ anomalies when preparing the manuscript of Goris et al. (2018) and found that less (or a different selection of) months do not provide higher correlations. This is because the seasonal $pCO_2^{sea}$ anomaly is chosen as a measure to capture the difference between models whose $pCO_2^{sea}$ seasonality is driven by variations in dissolved inorganic carbon (driven by mixed layer depth and biological production) and models whose $pCO_2^{sea}$ seasonality is driven by variations in sea surface temperature. As the here considered models have different timings for their peak in biological production (ranging from May to July) and as seasonal warming and biological production is not in phase (the peak in seasonal warming occurs in August for the here considered models), it is necessary to at least cover the months from May to August to capture the different seasonal drivers at play. However, further investigation revealed that seasonal warming is a dominant driver until the month of October. We have added a similar explanation in our revised manuscript.

-This study presented several predictors including the two from Goris et al. (2018). As shown in Fig. 8, each predictor provides a different estimate of the constrained range of future North Atlantic carbon uptake. Which estimate is more plausible?

**Answer 3:** We thank the reviewer for this very interesting question. We would like to first point out that our preprint states that "all newly constrained values for the future North Atlantic $C_{ant*}$ uptake are consistent which each other, i.e. the uncertainties around the constrained mean values are large enough for the solutions to not contradict each other" (lines 410ff). Accordingly, all estimates can be true at the same time. When it comes to the mechanisms that we are using to constrain the future North Atlantic $C_{ant*}$ uptake, it is our understanding that both emerging mechanisms are not completely independent in determining the $C_{ant*}$-uptake strength in the North Atlantic and they should not be viewed as two separate constraints. That is (i) the AMOC strength in the upper 500m drives the cold waters and the productivity levels in the high latitude North Atlantic; concurrently, (ii) the strength of the lower limb AMOC, which relates to the strength of upper limb AMOC, drives the effectiveness of surface-to-deep $C_{ant*}$ transport. For both upper ocean and deep ocean constraints, the AMOC-observations come with lower observational uncertainty, yet they represent a purely physical constraint such that we consider the biogeochemical constraints as more closely related to the North Atlantic $C_{ant*}$ uptake and hence more plausible. This is reflected in the fact that they also offer higher correlations with the North Atlantic $C_{ant*}$ uptake when compared to the AMOC-constraints in the same ocean depth-range. A lower observational uncertainty in the biogeochemical constraints would hence be of high value. We have added a similar explanation to our revised manuscript.

-How are the uncertainty range of the predictand $C_{ant*}$-uptake in Fig. 1, 8 and Table 1 calculated? I suppose they should be determined by the cross points of the linear regression line and the vertical lines of the observational uncertainty, but apparently it is not the case as shown in Fig. 1(c and e) and Fig. 8.

**Answer 4:** Indeed, this is not the case. We had noted in lines 401ff that "For details of the method that we utilise to calculate the unconstrained and observationally constrained estimates of the future North Atlantic $C_{ant*}$ uptake, the reader is referred to Bourgeois et al. (2022)." However, we understand that this statement is appearing too late in the manuscript and that a short introduction of the method would be helpful to the reader. Our method of estimating the constrained estimate follows the original approach of Cox et al. (2013). Here, the unconstrained estimate is given by the model mean and its uncertainty by the multi-model standard deviation. Assuming that all models are equally likely to simulate the true future

North Atlantic $C_{ant*}$ uptake and are sampled from a Gaussian distribution, a probability density function (PDF) can be calculated for the unconstrained estimate using model mean and standard deviation. Similarly, a PDF of the observational estimate and of the linear regression between predictor and predicant is established. For the observationally constrained future North Atlantic $C_{ant*}$ uptake, a conditional PDF is calculated by integrating over the product of the PDF of the observational estimate and the PDF of the linear regression. The observationally constrained estimate equals the expected value of the conditional PDF and the uncertainty of the estimate is given by its standard deviation. We have added a similar short introduction of the method to our revised manuscript, and, within the revised manuscript, we refer to this short introduction in the captions of Figures 1 and 6 as well as Table 1.

- It is not very clear how to interpret Fig. 5 and Fig. 7. The results in the two figures seem to contradict each other, both upper ocean (Fig. 5) and deeper ocean (Fig. 7) have high correlations. How to combine the information? In addition, L467-469: "…the deep ocean southward volume transport between 700m-4700m at 26N." Is this statement based on Fig. 7? This figure shows the 700m-5300m and 21N has reached the largest correlations.

**Answer 5:** We refer to Answer 3 and add that the strength of the northward mass transport with the AMOC (i.e., its upper cell or upper limb) is highly related to the strength of the southward mass transport with the AMOC (i.e., its lower cell or lower limb). Specifically, the upper branch of the AMOC transports warm waters from the low latitude to the high latitude North Atlantic, thereby releasing heat to the atmosphere (e.g., Rhein et al., 2011). Upon losing its heat, the water becomes denser and sinks. This densification links the warm, surface limb with the cold, deep return limb at regions of deep convection in the Nordic and Labrador Seas. For the Atlantic north of 26°N, volume conservation dictates that, for constant sea level, the net northward flow of upper waters balances the southward flow of deeper waters with a tolerance of 1Sv (McCarthy et al., 2015) such that there is a direct link between upper and lower cell of the AMOC. Though we are not considering the exact boundaries of the upper and lower limb, we have made an additional Figure (Figure R1, below) showing that the tight connection between upper and lower limb also holds for the here considered depth ranges. We have added a similar explanation to our revised manuscript.

The statement in L467-469 is based on two considerations: (1) While Figure 7 indicates that the southward transport between 700m-5300m reaches the highest correlations with the future North Atlantic $C_{ant*}$ uptake, the amount of $C_{ant*}$ that can be transported below 4700m is negligible. (2) We considered 26N instead of 21N as an observational constraint is available at 26N, while this is not the case at 21N. While we had denoted both considerations at different places in the preprint, we failed to summarise this more prominently. We have added a more prominent summary in our revised manuscript.

[Figure]

*Figure R1:* *Scatter plot of simulated mass transport at 26N in the upper 500m versus simulated mass transport at 26N between 700 and 4700m. The associated correlation coefficient is indicated in the upper left, the considered models and their color-coding is denoted in the legend.*

**Minor comments:**

-The information of figures are incomplete. I would suggest the authors to ensure all the figures are more or less self-explainable.

**Answer 6:** We thank the reviewer for pointing this out. As answer to this comment, we have re-done Figure 3 (see Answer 7), added units and expanded the captions for Figures 4 and 6 (see Answer 8) and presented the models in colours in Figure 8 (see Answer 9).

Fig. 3: the titles of x-axis and y-axis are missing, the y-axis' title is relative easy to guess, but the x-axis is not so straightforward. The readers need to check back and forth of the context to figure out that it should be number of iterations.

**Answer 7:** We apologize for overlooking this and thank the reviewer for pointing this out. Considering the previous comment of the reviewer, we decided to re-do Fig. 3 (depicted below) so that it hopefully is now self-explanatory and easier to understand.

[Figure]

**Figure 3:** *Iteration (population) versus cross-correlations for our application of the genetic algorithm. The correlation coefficients are calculated between contemporary values of the predictor within the best-performing regions identified by the genetic algorithm (per iteration) and future values of the predictand. For each shape (rectangle, ellipse, cuboid, and ellipsoid), three different applications of the genetic algorithm have been carried out based on different area or volume constraint (marked in blue, red and black). The green lines illustrate the cross-correlation without regional optimisation of the predictor.*

Fig. 4: the unit of the presented variable is missing on both plots and in the figure caption. Why are the color shadings much lighter in Fig. 4c-d than in the Fig. 4b, as they are presenting the same quantity? The same question is also for Fig. 6c-f.

**Answer 8:** We have added the units of the presented variables to our revised Figures 4 and 6. Colour shading in Fig. 4c-d as well as in Fig. 6c-d appear lighter than in Fig. 4b und Fig. 6b, respectively, as we had added a transparency of 70% such that the optimal regions are easier to identify. In our revised Figures 4 and 6, we have added the same transparency to all panels such that the colour shadings look the same in all sub-figures and do not lead to confusion.

Fig. 8: as specific model like CESM1-BGC is mentioned to perform well in L413-414, and more information and comparison can be made if the authors present the models with colors as in Fig. 1.

**Answer 9:** We followed the suggestion of the reviewer and revised Figure 8 to present the models in the same colours as in Figure 1.

-Abstract L3: "A previous study…" needs to add the reference paper citation so that the readers get the context. From reading the main text, I guess this study refers to Goris et al. (2018).

**Answer 10:** We thank the reviewer for pointing this out. The guidance for abstracts from GMD reads that "Reference citations should not be included in this section, unless urgently required". We interpret the reviewer's comments such that the reference is urgently needed and have added it to our revised manuscript.

-Abstract L5: "…winter $pCO_2^{sea}$ – anomaly…", but the previous paper (Goris et al., 2018) suggested the $pCO_2^{sea}$ anomaly in summer (May to October) NOT winter (November to April). As the winter and summer are taken actually half a year in this study, respectively, I guess the counterpart season should be with the same magnitude of correlation but a reversed sign. So I am quite confused that this study based on winter months and the previous study based on summer months get exactly the same correlations as shown in Fig. 1c.

**Answer 11:** This related to the fact that Goris et al. (2018) show the negative mean summer $pCO_2^{sea}$-anomaly in their Fig. 10 in order to be able to depict positive correlations. Here, the negative mean summer $pCO_2^{sea}$-anomaly equals the mean winter $pCO_2^{sea}$-anomaly (see Answer 2). We regret not having expanded on this in the manuscript and have added an explanation about the relation between negative mean summer $pCO_2^{sea}$-anomaly and mean winter $pCO_2^{sea}$-anomaly to our revised manuscript to avoid confusion. As this explanation is not fitting for the abstract, we have changed the wording in the abstract to "seasonal $pCO_2^{sea}$ anomaly" to avoid confusion.

-Some relevant details need to be described in this paper, so that the readers don't need to refer to Goris et al. (2018) all the time. For instance: how is the $pCO_2^{sea}$ anomaly defined, is it relative to the annual mean or long-term specific season mean? Which time periods are 1990s, 1997s, and 2090s?

**Answer 12:** Here, the 1990s are defined as an average over the years 1990-1999, the 2090s as an average over the years 2090-2090s and the 1997s as an average over the years 1997-2007. The last time-frame has been chosen as one of the utilised observation-based products is normalized to the year 2002. The mean winter $pCO_2^{sea}$-anomaly is defined to be the averaged November to April $pCO_2^{sea}$-values relative to the mean annual $pCO_2^{sea}$-values.

We have added explanations about the terms in question to our revised manuscript to avoid confusion.

-L350, 354, 363, Figs. S01, Figs. S03 and S04 are inconsistent with the figure numbering in the supplementary.

**Answer 13:** We have corrected this in our revised manuscript.

-L411: "…consistent which…" -> "…consistent with…"

**Answer 14:** We have corrected this in our revised manuscript.

-L487: "…averaged aver…" -> "…averaged over…"

**Answer 15:** We have corrected this in our revised manuscript.

**References:**

Cox, P., Pearson, D., Booth, B., Friedlingstein, P., Huntingford, C., Jones, C. D. and Luke, C. M.: Sensitivity of tropical carbon to climate change constrained by carbon dioxide variability. *Nature* 494, 341–344, https://doi.org/10.1038/nature11882, 2013.

McCarthy, G. D., Smeed, D. A., Johns, W. E.,Frajka-Williams, E., Moat, B.I., Rayner, D., Baringer, M.O., Meinen, C.S., Collins, J., Bryden, H.L.: Measuring the Atlantic Meridional Overturning Circulation at 26°N, Progress in Oceanography, Volume 130, 2015, 91-111, https://doi.org/10.1016/j.pocean.2014.10.006., 2015.

Mystakidis, S., Seneviratne, S. I., Gruber, N., and Davin, E. L.: Hydrological and biogeochemical constraints on terrestrial carbon cycle feedbacks, Environmental Research Letters, 12, 014 009, https://doi.org/10.1088/1748-9326/12/1/014009, 2017.

Qu, X., Hall, A., DeAngelis, A. M., Zelinka, M. D., Klein, S. A., Su, H., Tian, B., and Zhai, C.: On the Emergent Constraints of Climate Sensitivity, Journal of Climate, 31, 863 − 875, https://doi.org/10.1175/JCLI-D-17-0482.1, 2018.

Rhein, M., Kieke, D., Hüttl-Kabus, S., Roessler, A., Mertens, C., Meissner, R., Klein, B., Böning, C. W., and Yashayaev, I.: Deep water formation, the subpolar gyre, and the meridional overturning circulation in the subpolar North Atlantic, Deep Sea Research Part II: Topical Studies in Oceanography, 58, 1819–1832, https://doi.org/https://doi.org/10.1016/j.dsr2.2010.10.061, 2011.

Selten, F. M., Bintanja, R., Vautard, R., and van den Hurk, B. J. J. M.: Future continental summer warming constrained by the present-day seasonal cycle of surface hydrology, Scientific Reports, 10, https://doi.org/10.1038/s41598-020-61721-9, 2020.

Tokarska, K. B., Stolpe, M. B., Sippel, S., Fischer, E. M., Smith, C. J., Lehner, F., and Knutti, R.: Past warming trend constrains future warming in CMIP6 models, Science Advances, 6, eaaz9549, https://doi.org/10.1126/sciadv.aaz9549, 2020.

---

## Author Comment (AC5)

Answer to Review #2, https://doi.org/10.5194/gmd-2022-152-RC2

**We thank the reviewer for the thoughtful review. Detailed point by point responses to the major and minor comments are given below, with reviewers´ comments in black and answers of the authors in blue.**

**Review #2:**

I'm surprised to see this paper in review for GMD as it does not obviously meet any of the journal's manuscript types. It seems like the direct utility of this work is thinking about how to guide observational strategies to constrain N. Atlantic carbon uptake. This is however a call for the editor.

**Answer 1:** We would like to thank the reviewer for his/her very constructive review which has been very insightful for us. Though this is indeed a call for the editor, we would like to explain our reasoning behind choosing GMD to also assist the editor: It is our opinion that our paper meets the manuscript type "Methods for assessment of models" as it describes a "novel way of comparing model results with observational data". Specifically, our application of the genetic algorithm to regionally optimise emergent constraint had a twofold goal:

(1) To isolate key processes driving the multi-model spread, thereby enhancing our understanding of these processes, and identifying potential dynamical inconsistencies within the model ensemble

(2) To provide key areas where a narrow observational uncertainty is crucial for constraining future projections

Yet, the reviewer´s comments have made us aware that our way of structuring the manuscript needs to be improved, such that the original intent of the manuscript is better conveyed. We would therefore like to submit a revised manuscript that re-structures the content of our manuscript along this two-fold goal. More details are provided in Answer 2.

Major comments.

I find the manuscript comes across a bit as a dump of all the work the authors have done in this area, and as such, I feel it would benefit from some curating. The manuscript seems to be doing all of the following:

1. Identifying specific regions where people should be making observations to constrain future N. Atlantic $CO_2$ uptake (and in doing so they refine existing published emergent constraints slightly).

2. Exploring how a genetic algorithm can be used to select the optimum area of observational sampling to constrain models.

3. Expanding on the mechanisms behind the emergent constrains that the authors have previously put forward.

4. Better understand which key processes are leading to uncertainty in projections of future N. Atlantic $CO_2$ uptake (which links quite closely to 3).

As it is written it is doing 1, suggesting that it is doing 2, and doing a bit of 3 and 4 around the edges. The editor will be able to provide guidance on which of these a GMD paper should be doing, but I would argue that 2, 3 or 4 done fully would make the most useful papers, while1 is useful for a very specific audience. As it stands 2, 3 and 4 are the less developed parts of this manuscript. Perhaps it is OK to do all of these

things, but if that is what is done, a much clearer structure needs to be imposed on the manuscript and introduction of what is being done and why, so that the reader knows what information they should be getting from each section, and can efficiently take what they need from it. My preference would be to be clear about what the manuscript is trying to achieve and focus the manuscript on that, bringing in the other bits perhaps only as part of the discussion.

**Answer 2:** We thank the reviewer for his/her insights. Of course, it has not been our intention to present a dump of all the work that we have done in this area but instead to present a thorough study around the topic of Answer 1, with the North Atlantic carbon uptake as a case study. However, the reviewer´s comments has made us realise that the structure of our manuscript (1) went too quickly into the topic of the North Atlantic carbon uptake and (2) did not provide enough context into what information each Section is describing and therefore the twofold goal described in Answer 1 got lost between results.

In line with the reviewer´s request, we therefore propose to re-structure and extend our manuscript, such that becomes clearer that our manuscript is about

**Applying a genetic algorithm on existing emergent constraint to**
**(a) identify key model dynamics for the emergent constraint and model inconsistencies around them**
**(b) provide key areas where a narrow observational uncertainty is crucial for constraining future projections**

We propose to do this by firstly adding more introductory sentences that specify the content of each Section and the meaning behind this. Moreover, we propose the following structural changes:

- For the **title**: we propose to change the title to "Regional optimisation of Emergent Constraints: A case study for the North Atlantic carbon uptake"
- For the **abstract**: instead of directly describing the topic of the North Atlantic carbon uptake, we propose to begin our abstract with the topic of Emergent Constraints, followed by a motivation for our regional optimisation and an explanation of its two-fold goal.
- For the **introduction**: we propose to extend our introduction to not only explain the goal of narrowing down observational uncertainty in key areas but also to isolate key processes driving the multi-model spread, thereby enhancing our understanding of these processes, and identifying potential dynamical and systematic inconsistencies within the model ensemble.
- For the section **"Emergent constraints of the North Atlantic future carbon uptake"** which is part of "Background and experimental design": We propose to modify this Section by first introducing the method of Emergent constraints, including its caveats related to averaging over large areas before introducing the emergent constraints of our case study.
- For the section "**Genetic algorithm and experimental set-up**" which is part of "Background and experimental design": We propose to expand the title of the Section to "Genetic algorithm and experimental set-up for the regional optimisation" and to add an explanation as to why different shapes and sizes have been chosen for the regional optimisation and why we consider this set-up to be beneficial.
- For the "**Results**"-Section: Here, our introductory sentences will explain that Section 3.1 describes the performance of the Genetic Algorithm in terms of (1) speed of convergence towards an optimal solution and (2) improvement of correlations when applying the optimal regions to the Emergent Constraints. We will re-organise the remainder of the "Results"-section around our twofold goal of

(1) isolating key processes driving the multi-model spread, thereby enhancing our understanding of these processes, and identifying potential dynamical inconsistencies within the model ensemble; (2) providing key areas where a narrowing down of observational uncertainty is crucial to constrain future projections. Therefore, we propose the following structure:

*3.2 Optimal regions of the winter $pCO_2^{sea}$-anomaly and associated new emergent constraints*
*Visualisation and description of the optimal areas and their associated new emergent constraints*

*3.2.1 Plausibility of the optimal areas for the winter $pCO_2^{sea}$–anomaly*
*Description of the plausibility of the optimal areas including a dynamical reasoning*

*3.2.2 Implications of the optimal areas of the winter $pCO_2^{sea}$-anomaly*
*Description of inconsistencies within the model ensemble and where a reduction of observational uncertainty would help to disentangle the inconsistencies*

*3.3 Optimal regions of the fractional $C_{ant*}$-storage and associated new emergent constraints*
*Visualisation and description of the optimal areas and their associated new emergent constraints*

*3.3.1 Plausibility of the optimal areas of the fractional $C_{ant*}$-storage*
*Description of the plausibility of the optimal areas including a dynamical reasoning*

*3.3.2 Implications of the optimal areas of the fractional $C_{ant*}$-storage*
*Description of inconsistencies within the model ensemble and where a reduction of observational uncertainty would help to disentangle the inconsistencies*

- We will add a **"Discussion"**-Section, where we will present our approach and the additional information that it can give on (i) structural model error and (ii) the plausibility of different emergent constraints and compare our approach to other studies about the plausibility of Emergent Constraints.
- For the "**Summary and conclusion**"-Section: we will summarise as to why different shapes and sizes have been chosen for the regional optimisation and why this set-up is beneficial for disentangling of structural error and would be beneficial also for follow-up studies. Moreover, we will present a summary of our results along the lines of our twofold goal and what has been archived for each of the goals.

Fundamentally I can't see any mistakes beyond that raised by the other reviewer. I would echo the other reviewer's comments about it being difficult to interpret some of the figures, and would add that the manuscript would benefit from some careful editing for readability.

**Answer 3:** We have followed the advice of the other reviewer with regards to our figures. We also will edit our manuscript carefully with the intent to increase readability.

Minor comments:

- Just a comment - I'm pleased to see the desire for mechanisms in emergent constraints!
  **Answer 4:** We thank the reviewer for this comment.
- The title does not make sense. "Gulf Stream and interior western boundary volume transport as key regions to constrain the future North Atlantic Carbon Uptake" Should it perhaps read "Gulf

Stream and interior western boundary as key regions to constrain the future North Atlantic Carbon Uptake"?

**Answer 5:** We thank the reviewer for pointing this out. In line with preparing a differently structured manuscript, we will change the title to "Regional optimisation of Emergent Constraints: A case study for the North Atlantic carbon uptake"

- It seems to me that the 'competition' described in section 2 might benefit from a more detailed diagram than Fig 2.

  **Answer 6:** We are unfortunately unsure about what the reviewer means by ´competition´ as this term does not appear in our manuscript or Figure 2. We are happy to add a more detailed diagram once we are certain what process the reviewer refers to.

- Line 27 refer --> referred

  **Answer 7:** We have corrected this in our revised manuscript.

- Line 37 "Despite many progresses" --> "Despite much progress"

  **Answer 8:** We have corrected this in our revised manuscript.

- Line 37: 'have not necessarily' – be specific have they or haven't they, or in what areas have they.

  **Answer 9:** We understand the wish for explicitness, and have therefore revised Line 37ff and included three additional examples on top of the already existing example of equilibrium climate sensitivity, such that our revised manuscript reads:

  "Despite much progress in climate modelling, model bias and uncertainty (i.e., spread across models) have not decreased for all simulated variables. Most prominently, the model-generation of CMIP6 reveals the highest model uncertainty in equilibrium climate sensitivity when compared to other CMIP model-generations (Meehl et al., 2020). Similarly, Tagliabue et al. (2022) found that the absolute uncertainty in projections of global ocean net primary productivity has increased from CMIP5 to CMIP6. Additionally, their study points out that this growth in uncertainty substantially differs at regional scale. Contrarily, Terhaar at al. (2021) identify that the model uncertainty in surface density in the Arctic has decreased in CMIP6-ESMs when compared to CMIP5, leading to a reduced inter-model range of the anthropogenic carbon uptake in the Arctic. This result is echoed by Bourgeois et al. (2022), who find a smaller CMIP6 than CMIP5 model-uncertainty in both the contemporary ocean stratification and the anthropogenic carbon uptake in the Southern Ocean between 30°S and 55°S. Yet, the combination of large data volume and partially high model uncertainty in CMIP6 makes a comprehensive evaluation of associated models and simulations highly challenging."

- Line 72: "could highly gain from" --> "could gain from"

  **Answer 10:** We have corrected this in our revised manuscript.

- Line 214: "we advice against" --> "we advise against"

  **Answer 11:** We have corrected this in our revised manuscript.

- Line 487 aver should be over

  **Answer 12:** We have corrected this in our revised manuscript.

**References:**

Bourgeois, T., Goris, N., Schwinger, J., and Tjiputra, J.: Stratification constrains future heat and carbon

uptake in the Southern Ocean between 30∘S and 55∘S, Nature Communications, 13, https://doi.org/10.1038/s41467-022-27979-5, 2022.

Meehl, G. A., Senior, C. A., Eyring, V., Flato, G., Lamarque, J.-F., Stouffer, R. J., Taylor, K. E., and Schlund, M.: Context for interpreting equilibrium climate sensitivity and transient climate response from the CMIP6 Earth system models, Science Advances, 6, https://doi.org/10.1126/sciadv.aba1981, 2020.

Tagliabue, A., Kwiatkowski, L., Bopp, L., Butenschön, M., Cheung, W., Lengaigne, M., Vialard, J.: Persistent Uncertainties in Ocean Net Primary Production Climate Change Projections at Regional Scales Raise Challenges for Assessing Impacts on Ecosystem Services, Frontiers in Climate, 3, https://doi.org/10.3389/fclim.2021.738224 , 2021.

Terhaar, J., Torres, O., Bourgeois, T., and Kwiatkowski, L.: Arctic Ocean acidification over the 21st century co-driven by anthropogenic carbon increases and freshening in the CMIP6 model ensemble, Biogeosciences, 18, 2221–2240, https://doi.org/10.5194/bg-18-2221-2021, 2021.

---

## Author Response (AR2)

**Answer to Editor**

Dear Riccardo Farneti,

thank you for your positive decision and accepting our revised manuscript for publication.

We have taken the minor comments as provided by Reviewer #3 into account and provided a final version of our manuscript.

All the best,

Nadine Goris (on behalf of all authors)

**Answer to Referee #3, Review #1**

We thank the reviewer for the positive and thoughtful review. Detailed point by point responses are given below, with reviewers´ comments in black and answers of the authors in blue. Line numbers refer to the article document with tracked changes

**Review #1:**

In this article, the authors build on their previous work to identify of predictors for constraining future anthropogenic carbon uptake in the North Atlantic ocean in CIMP6 models. They propose and evaluate a novel application of a genetic method to regionally-optimize two previously identified "emergent constraints": the winter $pCO_2$ anomaly and the fraction of anthropogenic carbon stored below 1000 m. The genetic algorithm pointed out to a region of higher sensitivity of the future carbon uptake to the two predictors around the Gulf Stream, suggesting a dynamic control of the low-latitude AMOC on carbon uptake. The article is both of methodological and scientific value and, similar to the other reviewers, I could not identify any significant flaw in the science. The authors also re-worked the manuscript thoroughly following the advice of the previous reviewers. My only criticism is that the manuscript is a bit densely written (such that the focus is a bit lost sometimes – although the authors have made a significant effort to guide the reader in the revised version) and there are some repetitive bits, e.g. sections 3.2.1 and 3.3.1. I leave to the discretion of the authors whether they want to consider some re-structuring/simplification, which may improve the reading, but I don't see this as a real obstacle for accepting the article. Other than that, I came across some potential minor issues (note that line numbers refer to the Tracked Changes article document).

**Answer 1:** We would like to thank the reviewer for his/her thoughtful and positive evaluation of our revised manuscript. Based on her/his comments, we have tried to remove repetitive bits and to simplify or extend our text when fitting (especially in Lines 103-105, Lines 109-111, Lines 117-120, Lines 128-133, Lines 416-426, Lines 485-494, Lines 566-573).

- Line 150: "defined as an average over years 2090-2090s". Don't think this is correct.
  **Answer 2:** Indeed, this has been a mistake. We corrected this to "defined as an average over the years 2090 to 2099" (Line 136).

- Line 159. Not sure about the meaning of "partly" here
  **Answer 3:** We have removed the word "partly" to not confuse the reader (Line 146).

- Line 168. "… and biological production ARE not in phase"
  **Answer 4:** We have corrected this accordingly (Line 155).

- Line 384. I would use spuriously instead of randomly here.
  **Answer 5:** We have corrected this (Line 353-354).

- Similarly, I am not convinced of the used of phrase "by chance" (L. 395 and 533). I don't think such high and consistent correlations can emerge purely by chance, instead they could emerge from misrepresentations of processes in the model, as you mention elsewhere.
  **Answer 6:** We have corrected this (Lines 361 and 500).

- Lines 643-645. Something wrong with the brackets in this sentence?
  **Answer 7:** Our "tracked changes"-document did use a line-break at the incorrect place such that line 643 cut the words "linearity assumption)" off, including the closing bracket. This is corrected now (Lines 489-490).

- Line 700. Should it be "better-performing" instead of "well-performing"?
  **Answer 8:** We have corrected this to "better-performing" (Line 640).